# Evolution of gene knockout strains of *E. coli* reveal regulatory architectures governed by metabolism

Douglas McCloskey[1,2], Sibei Xu[1], Troy E. Sandberg [1], Elizabeth Brunk[1], Ying Hefner[1], Richard Szubin[1], Adam M. Feist [1,2] & Bernhard O. Palsson [1,2]

Biological regulatory network architectures are multi-scale in their function and can adaptively acquire new functions. Gene knockout (KO) experiments provide an established experimental approach not just for studying gene function, but also for unraveling regulatory networks in which a gene and its gene product are involved. Here we study the regulatory architecture of *Escherichia coli* K-12 MG1655 by applying adaptive laboratory evolution (ALE) to metabolic gene KO strains. Multi-omic analysis reveal a common overall schema describing the process of adaptation whereby perturbations in metabolite concentrations lead regulatory networks to produce suboptimal states, whose function is subsequently altered and re-optimized through acquisition of mutations during ALE. These results indicate that metabolite levels, through metabolite-transcription factor interactions, have a dominant role in determining the function of a multi-scale regulatory architecture that has been molded by evolution.

[1] Department of Bioengineering, University of California–San Diego, La Jolla, CA 92093, USA. [2] Novo Nordisk Foundation Center for Biosustainability, Technical University of Denmark, 2800 Lyngby, Denmark. Correspondence and requests for materials should be addressed to B.O.P. (email: bpalsson@ucsd.edu)

Biological response to gene loss can be evaluated on multiple time-scales. The immediate response to genetic perturbation is studied by measuring an organism's phenotypic response to a gene knockout (KO)[1–4]. For example, entire KO strain collections have been generated and used to define essential genes[5–8]. Besides assessing gene function, gene knockouts can be studied at the systems level through the integration of multi-omics data sets (i.e., metabolomics, fluxomics or network reaction rates, proteomics, and transcriptomics) to better understand the regulatory architecture that relies on the gene product. For example, it has been found that perturbations to the metabolic network are rapidly compensated for by flux re-routing caused by adjustments made at the regulatory level that re-tune enzyme level[1, 9]. Specifically, these studies found that regulatory changes (and in particular, changes in metabolite levels) occurred in proximity to the network lesion that a gene KO created. However, the extent to which distant regulatory changes relative to the location of the network lesion occurred was not discussed[1, 9]. In addition, the adaptive consequences of gene loss were not investigated.

The adaptive response to genetic perturbation is studied by measuring changes in physiological function after perturbation and during adaptation[10–12], and then characterizing the mutations that are required for the organism to regain the ability to grow optimally under the given conditions[13–26]. For example, it has been shown in bacteria and yeast that the likelihood of accumulating compensatory mutations is a function of the fitness cost of the KO[22–24]. Importantly, compensatory mutations often require the rewiring of existing regulatory networks to regain fitness, thus revealing the role of the lost gene in the regulatory architecture of the biological system[26]. Despite the potential to reveal novel insights into the regulatory architecture, to the best of our knowledge, a comprehensive systematic study looking at the rewiring of the regulatory network in response to gene loss has not been performed.

Previous work implemented a novel experimental design that involved gene knockouts (KOs) and adaptive laboratory evolution (ALE) in a pre-evolved *Escherichia coli* K-12 MG1655 strain (Fig. 1) to reveal detailed and mechanistic KO-specific adaptive responses to the loss of a gene[27–30]. Here, bioinformatics were implemented to reveal commonalities of how biological systems and specifically regulatory networks respond and adapt to gene KO at a systems level. First, the experimental design was confirmed through control evolutions of the pre-evolved strain. Second, multivariate statistical data decomposition methods found that the dominant modes of the data involved the drive towards regaining optimal fitness, while independent replicate evolutions revealed diversity in the adaptive paths selected in pursuit of optimal fitness. In this context, "optimal" indicates the biochemical state that allows for the maximal growth rate that the organism can achieve given the current environmental and genetic conditions. Third, biochemical pathway integration with multi-omic data sets revealed a common model of adaptive evolution. In this schema, network perturbation from gene KO altered metabolic flux, leading to perturbations in metabolite concentrations, which in turn triggered regulatory network responses altering gene expression. Gene expression responses were subsequently modified through mutations selected for during adaptation that improved fitness via ameliorated metabolic flux.

## Results

### Evolution experiment implementation.
A wild-type *E. coli* K-12 MG1655 strain previously evolved under glucose minimal media at 37 °C[31] (denoted as "Ref") was used as the starting strain in order to isolate biological changes caused by adaption to the loss of a gene product from those caused by adaption to the growth conditions of the experiment (Fig. 1e). Ref was a non-mutator strain and had the fewest number of mutations among the replicate adaptive laboratory evolution (ALE) endpoints generated.

Perturbations consisting of five separate metabolic gene KOs that were predicted to result in large metabolic rearrangements based on computational metabolic network analysis (see Methods, Supplementary Data 1) were implemented in Ref. Genes (see Methods) encoding enzymes for the reactions of GND (*gnd*, 6-phosphogluconate dehydrogenase), GLCptspp (genes *ptsH*, *ptsI*, and *crr* corresponding to enzymes HPr, EI, and EIIA, respectively), SUCDi (genes *sucA*, *sucB*, *sucC*, and *sucD* corresponding to the enzyme succinate dehydrogenase), TPI (*tpiA*, triosephosphate isomerase), and PGI (*pgi*, phospho-glucose isomerase) were removed to generate strains uGnd, uPtsHIcrr, uSdhCB, uTpiA, and uPgi, respectively (denoted "unevolved knockout strains" or "uKO"). GND generates D-ribulose-5-phosphate (ru5p-D), which is used in nucleotide biosynthesis, and re-charges NADPH, which is used for biosynthesis, in the final step of the oxidative Pentose Phosphate Pathway (oxPPP). *ptsH*, *ptsI*, and *crr* are primary components of the phosphotransferase system (PTS), which is the primary route for carbon import in *E. coli*, and aids in conserving energy by utilizing phosphoenolpyruvate (pep) to phosphorylate glucose instead of ATP. SUCDi couples the TCA cycle to respiration by charging and donating quinones to the electron transport chain (ETC) via Complex II. TPI avoids bifurcation of lower glycolysis by isomerizing dihydroxyacetone phosphate (dhap) to glyceraldehyde-3-phosphate (g3p) for subsequent enzymatic convert to pyruvate (pyr) via upper glycolysis. PGI converts glucose 6-phosphate (g6p) to fructose 6-phosphate (f6p) in the first committed step through upper glycolysis, thus controlling the flux split between the oxPPP and upper glycolysis.

Replicates of the five knockout strains, as well as Ref, were simultaneously evolved on glucose minimal media at 37 °C in an automated ALE platform[31, 32] denoted "evolved knockout strains" or "eKOi" where i denotes the replicate number. The number of replicate endpoints were the following: 2 for "evolved reference strain" (denoted eRef), 3 for eGnd, 4 for ePtsHIcrr, 3 for eSdhCB, 4 for eTpiA, and 8 for ePgi. Intracellular metabolite levels, gene expression levels, flux levels, and mutations (i.e., system components) were measured for the ref, uKO, and eKO strains during exponential growth. Intracellular metabolite levels consisted of close to 100 absolute and relative quantitative amounts of metabolites from glycolysis, the pentose phosphate pathway, the TCA cycle, energy and redox metabolism, cofactors, nucleotide metabolism, and amino acid metabolism[33, 34]. Gene expression levels consisted of relative fold changes from global RNA sequencing. Flux levels consisted of absolute intracellular flux values computed by metabolic flux analysis (MFA) using a genome-scale model from 13C isotope-labeling experiments[35, 36]. Mutations consisted of DNA resequencing mapped onto the reference *E. coli* K-12 MG1655 genome.

### Reference strain evolution confirmed the experimental design.
An insignificant fitness change and the fewest number of network changes were found in eRef strains compared to all eKO strains (Fig. 1e). The average numbers of significant component changes per eRef replicate at the metabolite, transcript, and flux levels were $2.0 \pm 0.0$, $35.0 \pm 5.7$, and $0.0 \pm 0.0$ (ave ± stdev, $n = 2$), respectively. These changes in systems components were far fewer than in any of the other eKO strains, where the minimum

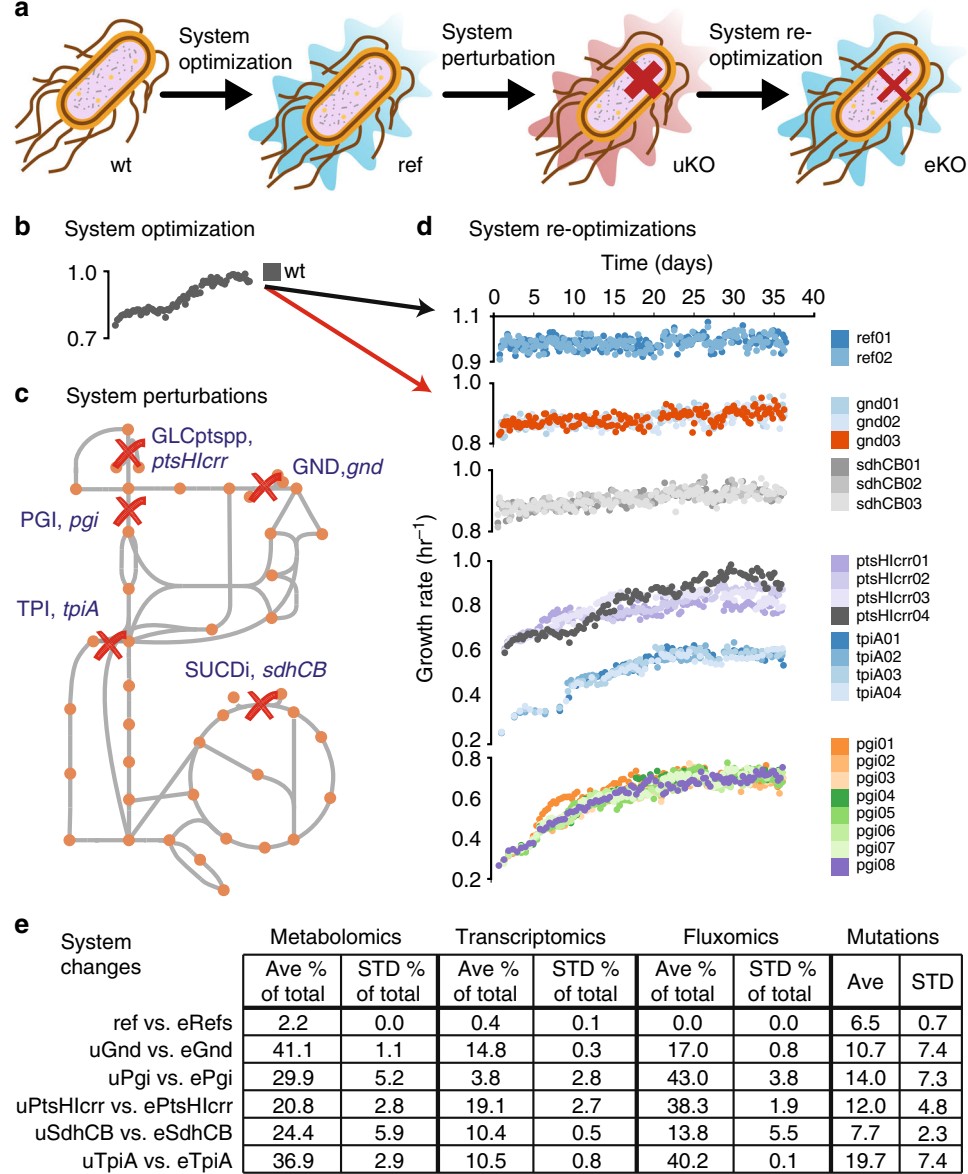

**Fig. 1** Evolution of knockout (KO) strains from a pre-evolved (i.e., optimized) wild-type strain. **a** Experimental design using adaptive laboratory evolution (ALE) and enzyme knockouts to investigate system re-optimization following major metabolic perturbations. **b** An isolated wild-type (wt) E. coli (MG1655 K-12) previously evolved on glucose minimal media at 37 °C[31] was used as the starting strain for knockouts of key metabolic genes and subsequent re-evolution, or systems re-optimization. **c** Reactions disabled by the enzyme knockouts included the phosphotransferase sugar import system (ptsHIcrr), phosphoglucose isomerase (pgi), 6-phosphogluconate dehydrogenase (gnd), triophosphate isomerase (tpi), and succinate dehydrogenase complex (sdhCB). **d** Adaptive laboratory evolution trajectories of the initial reference knockout and evolved knockout lineages. **e** Counts of significantly different system components found for each evolved knockout relative to the unevolved knockout. Counts of metabolomic, transcriptomic, and fluxomic data are given as the average and standard deviation of the percent of significant features compared to all features measured for the lineage; counts for mutations are given as the average and standard deviation of the number of significant features (see Methods for criteria for significance)

number of corresponding changes were 19, 341, and 158 (the average number of corresponding changes were $27.7 \pm 7.7$, $1051.6 \pm 513.7$, and $307.9 \pm 123.2$ (ave ± stdev, $n = 24$)). The average number of mutations per eRef replicate was also the lowest of all lineages, and were primarily found in cell wall biosynthesis genes. The average number of mutations per eRef was $6.5 \pm 0.7$, while the average number of mutations per all other eKO strains was $12.8 \pm 4.5$ (ave ± stdev). Overall, these findings demonstrated that the use of a pre-evolved strain minimized the number of confounding component changes.

**Evolution to optimal fitness was captured by the data**. Multivariate statistical analysis was performed on the data sets generated. Partial least squares discriminatory analysis (PLS-DA) revealed that the primary adaptive response to the gene KO involved a drive towards recovery of the optimal state (i.e., system re-optimization), followed by a secondary adaptive response that described unique alternate states that could be found at the newly evolved state. For almost all cases analyzed, the first most explanatory mode of PLS-DA (Fig. 2) separated Ref and eKO strains from the uKO strain (74% of eKOs from all data types and lineages, see Methods). This result indicated that the primary

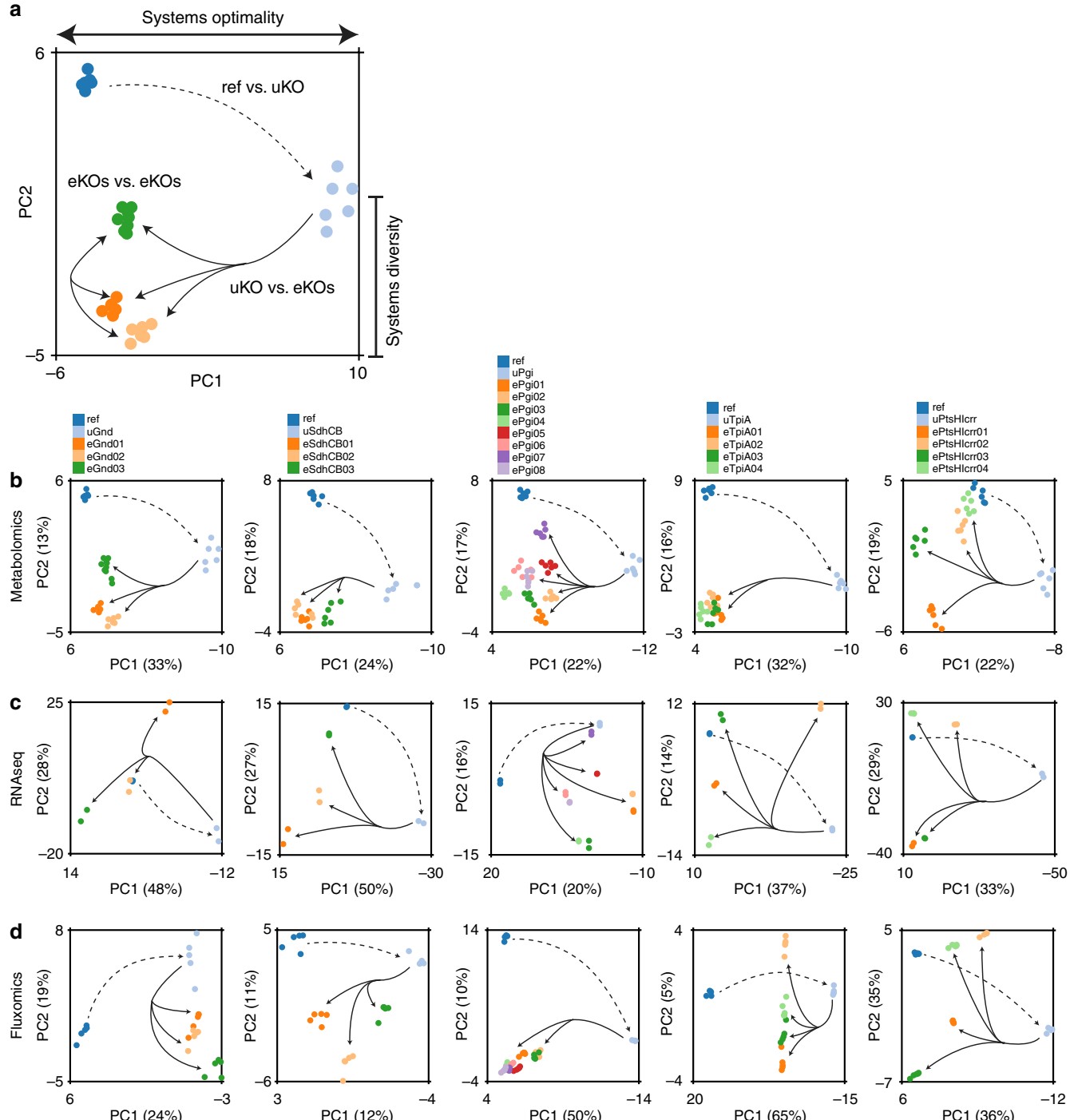

**Fig. 2** A multivariate analysis of biological network components as represented by different omics data types. **a** Partial least squares discriminatory analysis (PLS-DA) revealed a common trend in the two most dominant components: the primary component (PC1) most often corresponded to a movement away from (dashed line) and back to (solid line) evolved optimal fitness (i.e., optimal system configuration), while the secondary component (PC2) most often corresponded to a diversity among evolved optimal fitness states of different lineages (i.e., optimal system configurations). PLS-DA scores plots of the reference strain, initial knockout, and evolved endpoints for each lineage for metabolomics (**b**), transcriptomics (**c**), and fluxomics (**d**) data. The strain lineages denoted on the top of **b** also refer to the corresponding graphs below in **c** and **d**. All of the KO lineages matched the trend described above in the metabolomics data, one eKO did not match the trend in four of the five KO lineages in the expression data (i.e., all but eSdhCB), and one or more eKO did not match the trend in each of the KO lineages in the fluxomics data (see Methods for thresholds)

mode of the data accounted for a dominant transition between the Reference state, perturbed state, and evolved fitness states (i.e., captured systems fitness properties). This result was also reflected in the system component profiles themselves where the majority of component levels were restored or partially restored to reference levels (Fig. 3). For almost all cases analyzed, the second most explanatory mode of PLS-DA separated the Ref and eKO strains. This result indicated that the secondary mode of the data accounted for alternate evolved states (i.e., capturing systems diversity, or a 'plateau' in the evolutionary landscape[37]). These

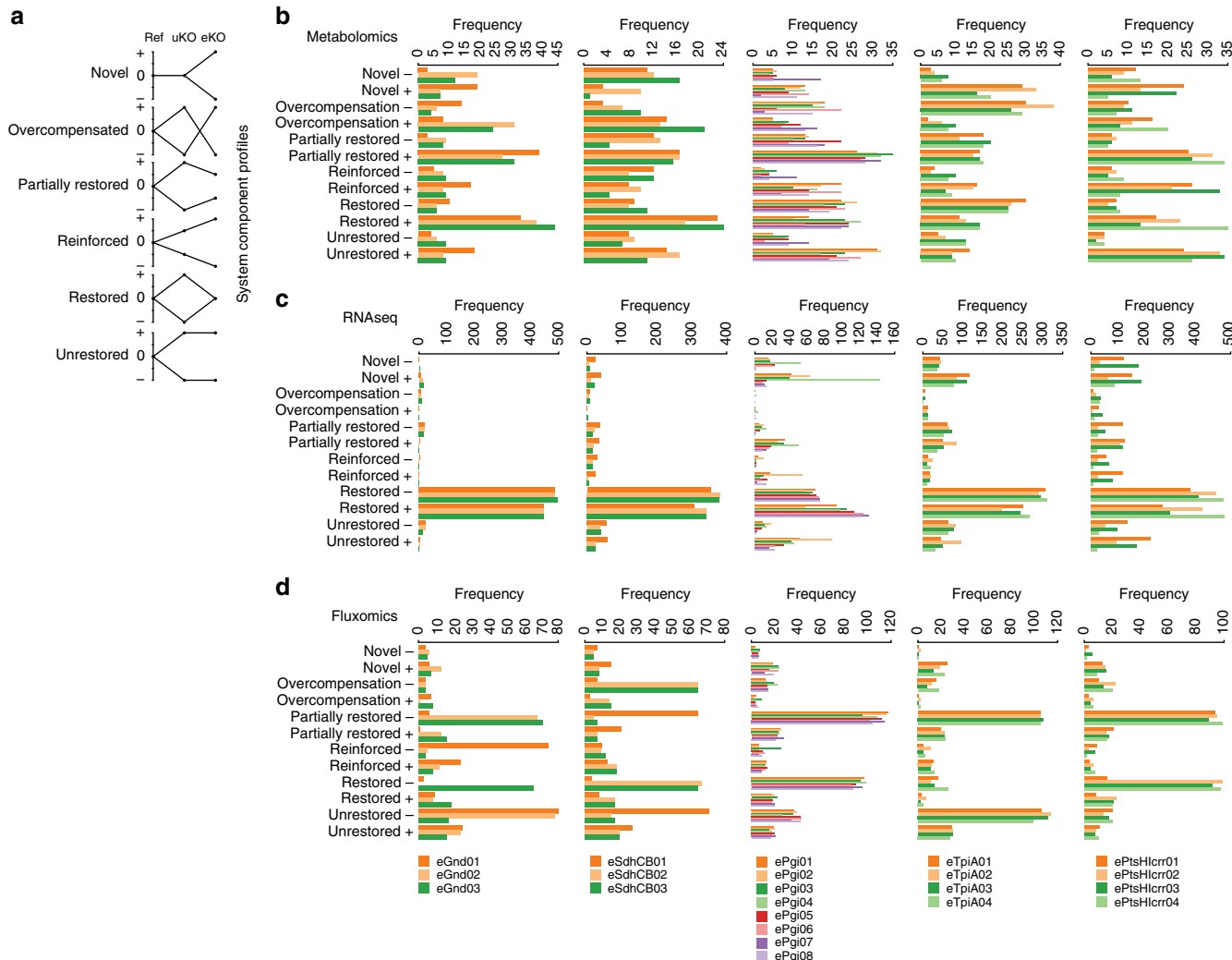

**Fig. 3** Classification of changes in omics data between the reference strain (Ref), the unevolved knockout strains (uKOs) and evolved knockout strains (eKOs). **a** Individual components were mapped onto six profiles according to their abundance in the Ref, uKOs, and eKOs in both positive and negative directions. The six profile types are shown and include novel, overcompensation, partially restored, reinforced, restored, and unrestored. The novel profile sought to categorize system components that were changed only as a result of adaptation. The restored and partially restored profiles sought to categorize system components that were initially perturbed to suboptimal levels post-KO. The overcompensation profile sought to categorize system components that overshot a restored profile to levels even lower/higher than those in Ref. The reinforced profile sought to categorize system components that needed a further increase or decrease after an initial perturbation post-KO to reach an optimal level. The unrestored profile sought to categorize system components that were immediately adjusted to an optimal level post-KO and required no further adjustments during adaptation. Metabolomics (**b**), transcriptomics (**c**), and fluxomics (**d**) data for each replicate of each KO lineage were binned into each of the six profiles (Pearson's R, R > 0.88). See section "Component profiles reveal systematic variations" for a discussion of trends that were found based on these six profiles

alternate states were a result of divergences in trajectory paths that led each replicate evolution towards a unique optimal state. This characteristic was further reflected in the unique distribution of component profiles between each of the eKOs.

**Component profiles reveal systematic variations.** In order to dissect the drive towards fitness (mode 1) and generation of diversity (mode 2) further, changes in each system component (i.e., metabolite, transcript, and flux level) between Ref, uKO, and eKO strains were grouped into six profiles (Fig. 3a, see Methods): novel, overcompensated, partially restored, reinforced, restored, and unrestored. The distribution between these six profiles for each component type are shown with horizontal bar charts in Fig. 3b–d. Several trends were found based on these six profiles.

First, the occurrence of profiles varied between omics data types. Overall, the metabolite levels were the most distributed

between the six profiles (i.e., had the least deviation). The ave ± stdev of the relative standard deviation (RSD) between profiles (n = 12, + and − directions for each of the six profiles) and across lineage (n = 22) was 39.9 ± 14.1, 132.1 ± 45.9, and 84.0 ± 12.7% for metabolites, transcripts, and fluxes, respectively. In contrast, the transcript levels were dominated by the restored profile, and flux levels were dominated by the restored and unrestored profile. For example, the *pgi* lineages had an ave ± stdev of restored profiles of 50.9 ± 5.0, 80.1 ± 8.3, and 66.9 ± 3.1% for metabolites, transcripts, and fluxes, respectively. The more even metabolite distribution compared to the transcript levels or flux levels indicated that the changes in metabolite levels were less constrained than the gene expression and fluxes.

Second, distribution amongst the profiles varied between KOs. The lineages with the greatest initial loss of fitness had a greater percentage of novel, overcompensated, reinforced, and unrestored profiles than the lineages with a smaller initial loss of fitness. This

difference was most evident for the transcript levels (ave ± stdev of 2.7 ± 0.4, 8.2 ± 3.7, 31.3 ± 23.4, 20.3 ± 10.9, and 18.6 ± 1.8%, and fitness change across evolution of 11.9 ± 3.9, 11.1 ± 2.9, 365.2 ± 20.0, 337.8 ± 73.8, and 244.3 ± 7.1% for the *gnd*, *sdhCB*, *pgi*, *ptsHIcrr*, and *tpiA* lineages; Pearson's $R = 0.94$, $P$-value < 0.017, Supplementary Fig. 1). This observation suggests that the larger the loss in fitness, the greater the number of Innovative (as opposed to restorative) network changes required to regain fitness. Future work with larger sample sizes will be needed to confirm this trend.

Third, the distribution amongst profiles also varied between evolved strain lineages. For example, the eight ePgi endpoints had varying levels of fitness (ave ± stdev of 0.68 ± 0.006, 0.61 ± 0.015, 0.65 ± 0.008, 0.72 ± 0.009, 0.64 ± 0.008, 0.69 ± 0.018, 0.67 ± 0.006, 0.69 ± 0.015 h$^{-1}$), and noticeable differences in the distribution of profiles among endpoints. This highlighted the biochemical differences in evolved network configurations during adaptation to overcome the perturbation.

Finally, a decoupling between degree of fitness change and degree of -omics data change was apparent. The *tpiA*, *pgi*, and *ptsHIcrr* lineages incurred the largest loss and recovery of fitness while the *gnd* and *sdhCB* lineages incurred only minimal changes in fitness. However, major changes in all -omics data measured between Ref and uKO and between uKO and eKO strains were found in all lineages (Figs. 2 and 3). Interestingly, major changes often occurred in common system components. Major changes could be traced to either perturbed metabolites that act as allosteric or transcriptional regulators (which is consistent with previous studies[38, 39]) or mutations that resulted in alterations to gene expression. The observation about commonly perturbed metabolite levels and mutations coupled with our previous three observations about the profile distributions indicated that changes in fitness and -omics data were independent, given that major alterations in gene expression and protein production could occur as a result of perturbations in relatively few key regulators.

The system component profiles were mapped to the biochemical network of *E. coli* and analyzed to develop a general framework for understanding evolution at the molecular level. It is important to highlight that the component profiles described above were used in all of the analyses presented below. The component profiles were assigned based on statistical criteria. They provided a unitless metric to compare and map multiple data types when quantitative relationships between data types have not been fully established. The component profiles also provided robustness by basing the analysis on change in values between states (i.e., ref, uKO, and eKO) instead of the absolute value found in any one state.

**Changed flux distribution was most prevalent during ALE.** Changes in pathway usage between the Ref, uKO, and eKO strains were calculated, and differences between the flux distribution in the uKO and eKO strains were grouped into changed flux distribution (i.e., the pathway usage was changed) or changed flux capacity (i.e., the same pathway was used but at a higher flux level, see Methods for extended definitions, Fig. 4a–d). Changed flux distribution was found to be more prevalent than changed flux capacity. Changed flux distribution was found to occur 55.6% of the time, while a change in flux capacity was found to occur 22.0% of the time across all perturbations and lineages (Fig. 4e). The remaining 22.4% of cases were unaffected.

For example, flux was initially re-routed through the Entner–Douderoff (ED) pathway in uGnd (Fig. 4f) in order to generate ribose through the non-oxidative Pentose Phosphate pathway (non-oxPPP). The ED pathway has a net yield of one ATP, NADH, and NADPH per molecule of glucose, whereas

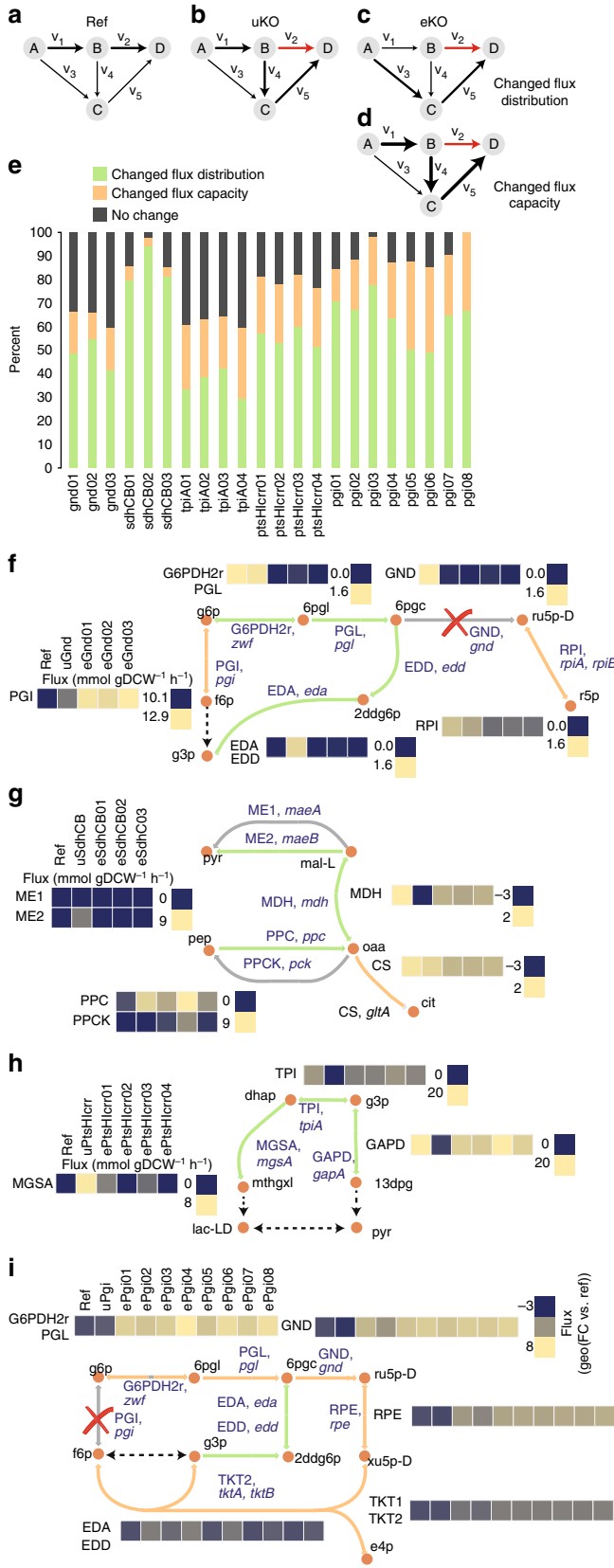

glycolysis has a net yield of two ATP and NADH[40]. Instead, the eGnd strains limited the use of the oxidative pentose phosphate pathway (oxPPP) and increased the flux capacity through the higher energy and redox equivalent producing pathway of glycolysis. Further examples are given in Fig. 4. These results

**Fig. 4** Suboptimal pathway usage limits allocation of carbon to biomass precursors. Toy network schematic of flux distribution in Ref (**a**) and in uKO (**b**). A reaction knockout is highlighted in red. The flux distribution in eKO could be categorized as **c** changed flux distribution (i.e., the pathway usage was changed) or **d** changed flux capacity (i.e., the same pathways was used but at a higher level, see Methods). Four examples of changed flux distribution and changed flux capacity for **f** gnd, **g** sdhCB, **h** ptsHIcrr, and **i** pgi lineages. **f** flux was initially re-routed through the ED pathway after removing the gnd gene The ED pathway has a net yield of one ATP, NADH, and NADPH per molecule of glucose, whereas glycolysis has a net yield of two ATP and NADH[40]. Instead, the evolved gnd endpoints limited the use of the PPP and increased the flux capacity through the higher energy and redox equivalent producing pathway of glycolysis. **g** flux was initially re-routed through the TCA cycle in uSdhCB by diverting flux through the anaplerotic reactions phosphoenolpyruvate carboxylase (PPC) and inverting the direction of flux through malate dehydrogenase (MDH). The eSdhCB re-inverted the direction of malate dehydrogenase towards production of nadh or quinone reduction, and downregulated flux through the rest of the TCA cycle. **h** A significant portion of flux was bifurcated between the methylglyoxal pathway and lower glycolysis in uPtsHIcrr in response to elevated levels of dihydroxyacetone phosphate (DHAP) and depletion of lower glycolytic intermediates that inhibit the activity of methylglyoxal synthase[114, 115]. The flux through the methylglyoxal pathway was essentially eliminated in endpoints 2 and 4, and significantly decreased in replicates 1 and 3, in order to utilize the less toxic and more energy and redox producing lower glycolytic pathway. **i** The abnormally high levels of flux directed through the oxidative Pentose Phosphate Pathway (oxPPP) in uPgi was initially re-routed through the ED pathway. Several evolved pgi endpoints retained the flux through the ED pathway to varying degrees, but most re-distributed flux through GND, and all increased the flux capacity through the non-oxidative Pentose Phosphate Pathway (non-oxPPP). Green and orange colored reaction lines in **f**–**i** correspond to the grouping of changed flux distribution or changed flux capacity shown in the bar plot in **h**. Color bars for all flux values are shown next to their corresponding reaction(s)

indicated that the initial flux distribution of the uKO strains following perturbation were often suboptimal, and required a change primarily in flux distribution and secondarily in flux capacity in order to restore fitness in the eKO strains.

**Perturbed metabolite levels triggered TRN responses in uKOs.** Transcriptional regulatory network (TRN) responses in uKOs that were associated with carbon metabolism, nitrogen metabolism, iron regulation, oxidative stress, DNA repair, and other stress responses that control the majority of known functions in *E. coli* were linked to corresponding changes in regulatory metabolite levels (see Methods). Perturbed metabolite levels were traced to known TRN responses[41–45] by mapping measured metabolite profiles to metabolite-activated transcription factors (TFs). The relationship (i.e., positive or negative) between a metabolite profile, a TF that interacts with the metabolite, and the expression profiles of the transcription units (TUs) regulated by the TF (see Methods, Fig. 5) were compared. Strong evidence (i.e., statistically significant gene expression pattern for genes that are regulated by a single TF, see Methods) for changed TF activation profiles (analogous to the system component profiles, Fig. 2) were identified for 75 TFs (Supplementary Data 2, Fig. 5). These included 7 global TFs (i.e., CRP, Fis, IHF, ArcA, Lrp, FNR, and HNS[46]) and 68 pathway-specific TFs (see Methods). The activation profiles of 15 TFs (which included the 7 global TFs and the 8 pathway-specific TFs ArgR, CpxR, Cra, Fur, NsrR, OxyR, PhoB, and TyrR) were changed across all lineages. The remaining 60 TFs appeared to be changed in a perturbation and lineage-specific manner.

Interestingly, TF activation and TF gene expression was not coincidental (ave ± std 5.4 ± 3.8, 4.1 ± 2.6, and 70.5 ± 6.1% agreement, disagreement, no significant change in expression profile per lineage, respectively, Supplementary Data 3). This result indicated that changed TF activation was mostly attributed to changed concentrations in their metabolic activators as opposed to changed TF gene expression levels. Similar observations have been made for sigma factors and the expression levels of sigma factor DNA binding operons in response to a key *rpoB* mutation, where alterations in the binding of the regulator subsequently altered gene expression of regulated operons[47]. For example, a changed CRP activation was found in all lineages due to elevated levels of cAMP in the uKOs[48]. CRP was not differentially expressed in any of the lineages, but restored cAMP levels were mirrored by restored gene expression of TUs solely regulated by CRP-cAMP (Supplementary Data 5). ArcA provided another example for global TF activation without a significant gene expression change. The restored activation profiles of ArcA[49] and several other iron–sulfur cluster homeostasis TFs found in all lineages could be linked to changes in TCA cycle intermediates as well as quinone pools (e.g., *gnd* and *sdhCB*). The ArcAB two-component system in particular modulates genes in response to changes in respiratory conditions that are communicated via the intermembrane quinone pools.

Pathway-specific TF activation was also identified in the uKOs. A change in activation of the PurR regulator was found in *pgi* and several other lineages due to changed levels of purine degradation products. Specifically, the *purR* dimer binds hypoxanthine and guanine, and regulates genes involved in purine metabolism[50–52]. The concentration profiles of hypoxanthine and/or guanine matched the expression profile for purR-target genes, while the expression profile for *purR* itself did not (Supplementary Data 4). In another example, the change in activation of TyrR in many lineages was found to be attributed to the change in levels of L-tyrosine and L-phenylalanine[53] (Supplementary Data 4). TyrR binds L-tyrosine and L-phenylalanine and modulates genes involved in aromatic amino acid production and transport. The component profile of L-tyrosine was found to match the expression of *aroF*. The component profile of L-tyrosine and *aroF* was also consistent with TyrR activation by L-tyrosine and regulation of *aroF* gene expression, which indicates that *aroF* gene expression was modulated by L-tyrosine levels via TyrR. Expression of *aroF* is controlled only by TyrR[54]. Another example of pathway-specific TF activation involved the use of small regulatory RNA. A sugar phosphate toxicity response was generated by abnormal elevations in glucose 6-phosphate (g6p) and an imbalance of the glycolytic intermediates in uPgi. SgrR is thought to bind hexose phosphates and induce the expression of the small RNA *sgrS*[55–57] (Fig. 5), which initiates the observed response. It was found that the metabolite concentration profiles matched *sgrS* expression profiles. *SgrS* transcriptionally regulates a number of genes that are involved in re-balancing glycolytic intermediates. One target of *sgrS* attenuation is *purR*, which explains the opposing *purR* expression profile compared to its TF activation profile described above. Interestingly, abnormal elevations of g6p and induction of SgrR and SgrR regulons were also found in ptsHIcrr. Additional examples are provided in Fig. 5.

The common perturbation of TFs by small molecules indicated that the majority of transcriptional changes observed may not be beneficial to fitness compensation, but a consequence of "hard-coded" regulatory circuits selected for through evolution that were triggered by perturbations to key metabolite regulators. Many of the hard-coded regulatory circuits were revealed through ALE.

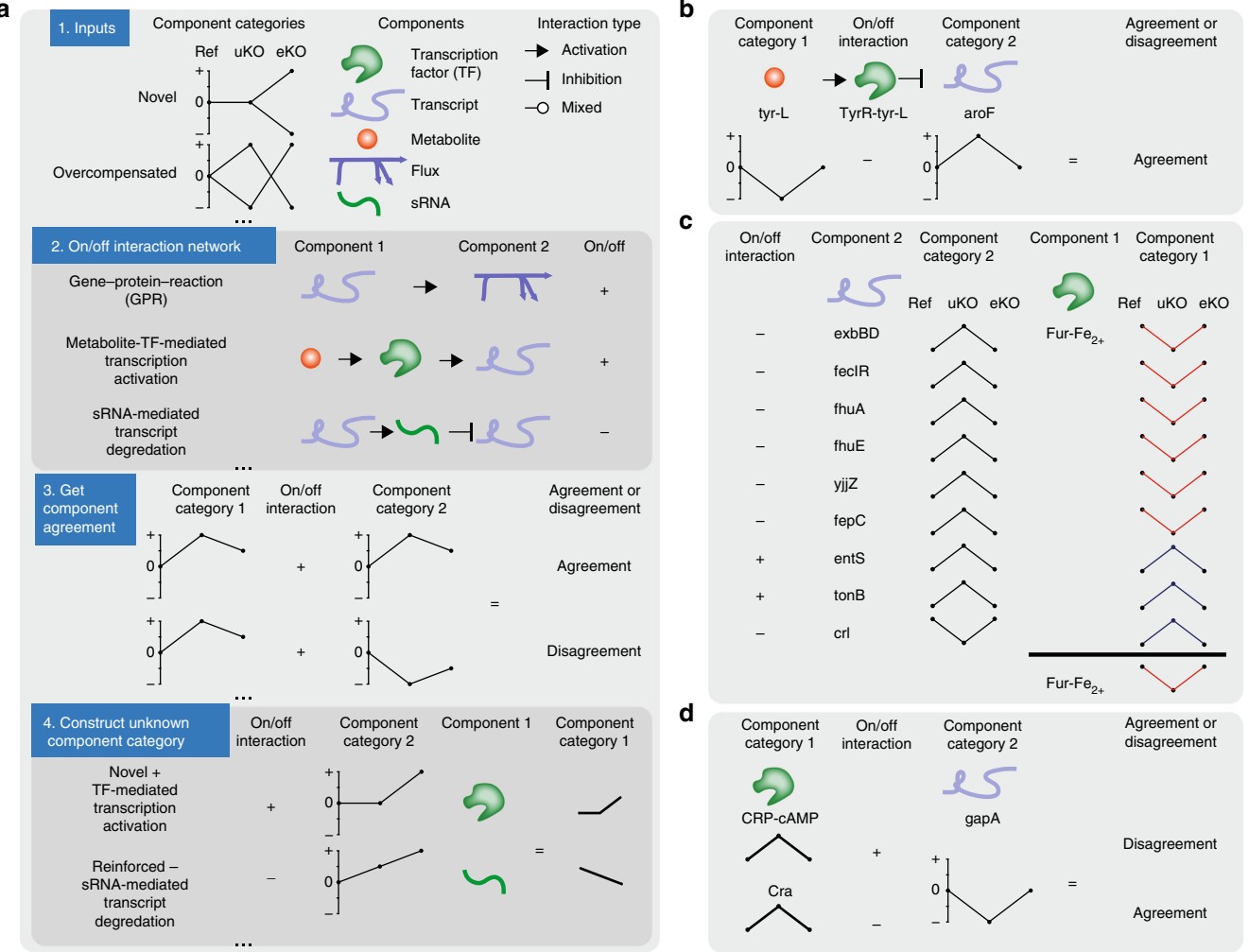

**Fig. 5** Mapping between network components and annotated regulation. **a** An algorithm for determining agreement and disagreement between system components categories and annotated biochemical pathways and regulation. 1. The algorithm inputs include the component profiles, the network components, and the network interactions. 2. An on/off Boolean interaction network that describes the biochemical and/or regulatory relationship between two components is constructed. 3. The component categories and on/off interaction between each component can then be determined. 4. For components that were not directly measured, a consensus category and confidence score can be determined. **b** Example of metabolite-mediated transcription factor activation between tyr-L, TyrR, and aroF[53]. **c** Example of an unresolved discrepancy involving Fur regulation. **d** Example of transcription factor hierarchy between cAMP-CRP and Cra

**Component profiles revealed competing layers of regulation.** Cells contain multiple levels of counteracting regulatory mechanisms that often overlap. For example, a relatively low agreement between changes in gene expression profiles and metabolic flux profiles (i.e., gene–protein–reaction association, GPR) within each lineage was found (Supplementary Data 2). Specifically, an average agreement of 27.5% (stdev = 17.4%, $n$ = 22) and average disagreement of 11.5% (stdev = 6.8%, $n$ = 22) was found. A similarly low agreement between types of literature-derived regulation were found (Supplementary Data 2). These findings are consistent with previous work and can be explained by the actions of multiple and competing layers of regulation[58, 59].

Competing levels of regulation can be measured through the disagreement between changes in system components and literature-derived networks of biomolecular interactions (Fig. 5). Disagreements were found to categorize into three main groups: (1) counteracting regulatory mechanisms, (2) evidence for inaccurate or incomplete knowledge of regulatory networks[60–63], and (3) changes to regulation introduced through fixed mutations. Evidence of competing layers of regulation for 89

regulators (i.e., any biological component that can affect a change in another component, e.g., TF or small-molecule) across 5887 regulated entities (i.e., any biological component that is subject to regulation, e.g., TU or enzyme) were found. Evidence of inaccurate or incomplete knowledge of the regulatory network in 38 regulators across 631 regulated entities were found (Supplementary Data 3). While it is infeasible to investigate each discrepancy here, specific examples are given that illustrate the above three mechanisms.

In an example of counteracting regulatory mechanisms, a hierarchy of TF control over gene expression was recapitulated. The activation profile of Fis[64–66] was found to conflict with its consensus activation profile of the *pyrD* promoter in all of the *pgi* lineages, whereas the PurR activation profile was found to agree with *pyrD* expression profile[52, 64, 65]. This indicated that *pyrD* expression was dominated by PurR regulation. In another example, a restored activation of *sgrS* found in the *pgi* lineages and a novel activation of *sgrS* found in the ptsHIcrr endpoints 1 and 3 negated the transcription factor regulation of sgrS target genes[67, 68]. In another example, the activation profile of cAMP-CRP was found to conflict with its consensus activation profile

on the *gapA* promoter in all of the *tpiA* lineages, whereas the Cra activation profile was found to agree with *gapA* expression profile (Fig. 5d)[69, 70]. cAMP-CRP and Cra bind upstream of the promoter region of *gapA*; CRP-cAMP promotes *gapA* transcription while Cra inhibits *gapA* transcription[69, 70]. This finding indicated that inhibition of *gapA* expression by Cra was dominant over the promotion of *gapA* expression by cAMP-CRP, as is consistent with recently reported data[71]. In another example, the activation profile of the TF Nac, which acts as a global regulator of nitrogen metabolism,[72] was found to conflict with its consensus activation profile for the expression of *gabP* on the *csiD* promoter in *tpiA* replicates 1 and 2. Expression of *gabP* is controlled by cAMP-CRP, CsiR, HNS, and Lrp[41, 73]. Only the activation profile of Lrp matched, indicating that the expression of *gabP* was dominated by Lrp in those two replicates. In another example, the transcription attenuation by UTP was found to dominate the regulation of *pyrLBI* operon by ppGpp[74, 75].

Unresolved discrepancies in regulatory annotations were found. The expression profiles of regulons that were controlled only by Fur[76–78] were found to be inconsistent. Specifically, the expression profiles for *entS*, *exbB*, *exbD*, *fecI*, *fepC*, *fepD*, *fhuA*, *fhuE*, *ryhB*, and *yjjZ*, conflicted with that of *crl* (Fig. 5c). The discrepancies indicated that another TF or transcriptional regulator is present that also controls the transcript levels of that gene or Fur can act as a dual regulator similar to entS[79]. In fact, *crl* has been shown to also be regulated by ArgR[45] and positively regulated by CsrA[80]. In addition, *yjjZ* has also been shown to be positively regulated by OxyR[81] and positively and negatively regulated by Fnr[43]. In another example, the *yeiP* gene was annotated to be regulated only by cAMP-crp[41, 70]. However, the expression profile of *yeiP* conflicted with the consensus activation profile of cAMP-crp across all lineages.

Discrepancies arising from changes to regulation introduced through mutation were also identified. For example, the *lon*-specific promoter is activated by GadX[41, 82, 83]. A mutation at the *lon*-specific promoter in the ePgi replicates 1-5 silenced the expression of *lon* thereby negating the regulation by GadX. This silencing directly affected the expression of colanic acid and biofilm producing operons that are controlled by RcsA and RcsAB[84]. The Lon protease degrades RcsA[85]. Further examples are given in more detail below.

These examples demonstrate the hierarchical and interconnected web of regulation found in the cell, and demonstrate how changes to one regulator can impact the regulation of biological components at multiple system levels. In addition, the examples given above indicated that the response of the uKO and eKOs recapitulated the effects of known regulation, but also revealed the effects of unknown or not fully characterized regulatory mechanisms. The latter provide suggestions for new experimental lines of inquiry.

**Mutations altered regulation and enzymatic function**. A large number of mutations were identified in the eKOs that changed the effects of global and pathway-specific regulators (discussed above) or targeted specific pathways or imbalances. In total, 673 mutations were found in the eKOs (Supplementary Data 5 and 6). The mutations were found to primarily be single nucleotide polymorphisms (SNPs, 66%), were primarily located in coding regions (48%), and were primarily associated with membrane proteins and transcription factors (27 and 29%, respectively). See Supplementary Data 5 and Fig. 6 for a detailed overview of all mutations found in the eKO strains. The reader is directed to McCloskey et al.[27–30] for further in depth characterizations of individual mutations discussed below.

Mutations selected during ALE changed many global regulators. For example, 17% of mutations affected regulators of carbon transport and metabolic processes that appear to offset the activation of operons induced by CRP-cAMP. These included mutations to *galR*, *malT*, and *crr* in the ePgi strains that appeared to negate repression of *galR* controlled operons. The mutations may give the evolved strains an additional route to import and catabolize glucose because the galactose importer also has the ability to import glucose albeit with lesser affinity than galactose. In addition, the mutation may have improved the fitness of the ePgi strains by increasing the availability of phosphoenolpyruvate (pep) for aromatic amino acid production. Interestingly, mutations in *galR* or at the *galR* operon in ePtsHIcrr02/04 and in eTpiA01/03 also resulted in the upregulation of GalR controlled genes. The prevalence of *galR* mutations may indicate that expression of the *gal* regulon may aid in increasing fitness when the ability to import glucose is impaired or the levels of pep are inadequate for aromatic amino acid production. Additional mutations that affected carbon transport processes included *ptsG*, *galR*, and *nagC* in the ePtsHIcrr strains, and *ptsG*, *galR*, and *nagA*, *nagC*, and *nagE* in the eTpiA strains.

A series of mutations were also identified that altered protein homeostasis networks, two-component systems, small RNA networks, and the sigma factor networks. These included mutations that altered the Lon protein homeostasis network in ePgi and the two-component system RcsA/RcsB in ePtsHIcrr that targeted pathways involved in cell motility, acid resistance, and cell wall biosynthesis. Mutations that altered the SPF small RNA networks, RpoC core RNA polymerase unit, and RpoD sigma factor networks in ePGI were found. Alterations to stress response systems that included SoxS/SoxR in *pgi* and PhoB/PhoR in *tpiA* involving oxidative stress and phosphate stress, respectively, were also found.

Mutations were also identified that changed the regulation of pathway-specific TFs. These occurred in a KO-specific manner, and appeared to optimize specific pathways at the regulatory level. For example, the expression of the methylglyoxal pathway in eTpiA strains were altered to more efficiently convert methylglyoxal to lactate through mutations that altered methylglyoxal detox pathway gene expression. These examples of global and pathway-specific regulatory shifts indicated that mutations that affect hubs in complex regulatory networks are common in adaptive evolution[37], and provide a fitness advantage by rewiring regulatory network responses that may no longer be optimal for fitness.

Rarer were mutations that introduced innovations that appeared to target-specific metabolite imbalances. For example, the levels of nadph, which is used to drive biosynthesis, was affected in many of the KOs. Mutations were found in the trans hydrogenases in several of the ePgi strains and in all of the eGnd strains to compensate for an overproduction and underproduction of NADPH, respectively. A mutation found in the active site of seven of the eight ePgi endpoints in isocitrate dehydrogenase appeared to alter cofactor specificity to allow for the use of nadh.

## Discussion

Taken together, the combination of study design, automated ALE, multi-omic data sets, and statistics and bioinformatics revealed common mechanisms of adaptation whereby imbalances in metabolite levels from altered fluxes triggered a multitude of network responses that were readjusted by mutations selected for during evolution (Fig. 7). The mutations that fixed during adaptation acted to rewire many existing hardwired responses and/or introduce novel network functions that addressed the imbalances that the initial KO lesion created. The findings of this

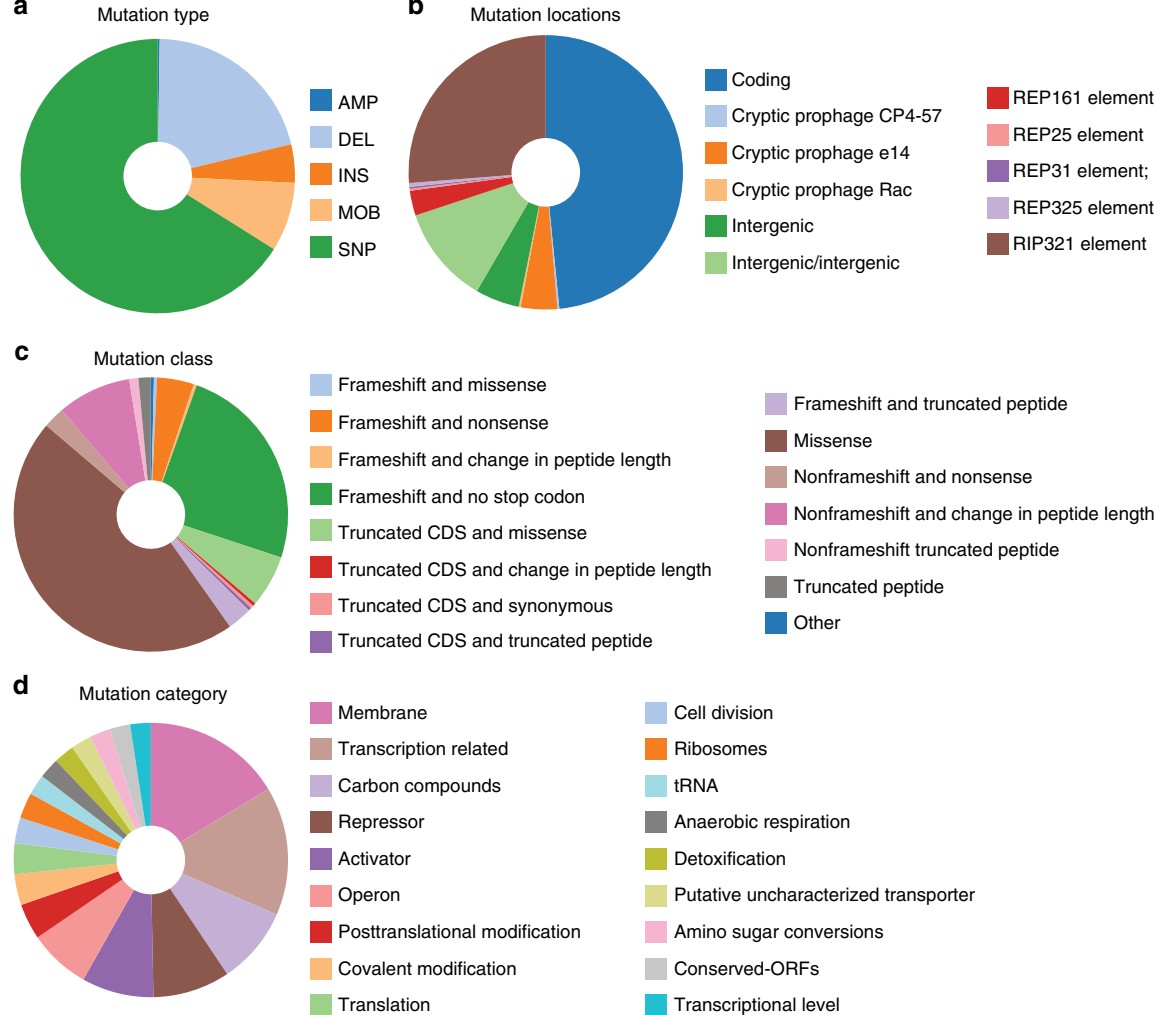

**Fig. 6** Overview of mutation statistics. See Supplementary Data 6 for detailed statistics of each category and categories not shown. **a** The type of mutation. Mutations include amplification (AMP), deletion (DEL), insertions (INS), mobile element aided insertions or deletions (MOB), single nucleotide polymorphism (SNP). **b** The location of the mutation. Locations include coding regions, regions associated with cryptic prophages, intergenic regions, regions two coding genes not classified as an intergenic region (intergenic/intergenic), and repetitive elements (REP or RIP). **c** The class of mutation. Classes include frameshifts, frameshifts resulted in a truncated CDS, missense, non-frameshifts, peptide truncations, and other unclassified mutations. **d** The functional or structural category of the mutated gene. Categories are based on the "parent class" as found in the EcoCyc database[103]

study represent a step towards developing a fundamental understanding of how cells mechanistically adapt to gene loss from a systems perspective that accounts for proximal and distal relationships in the metabolic and regulatory network. Novel mechanisms and inconsistencies, revealed through adaptation, between measurement and known regulatory mechanisms identified in the case studies present opportunities for future discovery (Supplementary Data 2 and 4). Specific avenues of exploration may include the effect of regulation acting on different timescales (i.e., transcriptional vs. allosteric regulation) or the effect of RNA and protein stability and degradation that were not addressed in this study.

## Methods

**Biological material**. A glucose, 37 °C, evolved *E. coli* derived from *E. coli* K-12 MG1655 (ATCC 700926)[31, 32] served as the starting strain. Lambda-red-mediated DNA mutagenesis[86] was used to create the knockout strains. Knockouts were confirmed by PCR and DNA resequencing. Genes gnd, ptsH, ptsI, crr, sdhC, sdhA, sdhD, sdhC, tpiA, and pgi encoding for the reactions of 6-phosphogluconate dehydrogenase (GND), phosphotransferase sugar import (GLCptspp), succinate dehydrogenase complex (SUCDi), triophosphate isomerase (TPI), and phosphoglucose isomerase (PGI) were removed. PPC was also deleted, but resulted in

an auxotrophy for asp-L, and was not included in the study. Genes aceE, aceF, zwf, and atpI-A encoding for the reactions of PDH, G6PDH2r, and ATPS4rpp could not be removed using the method of Datsenko et. al[86]. All cultures were grown in unlabeled or labeled glucose M9 minimal media[87] with trace elements[88] at 25 mL of working volume in a 50 mL autoclaved tube. The cultures were maintained at 37 °C on a heat block and aerated using magnetics.

**Materials and reagents**. Uniformly labeled $^{13}$C glucose and 1-$^{13}$C glucose were from Cambridge Isotope Laboratories, Inc. (Tewksbury, MA). Unlabeled glucose and other reagents were from Sigma-Aldrich (St. Louis, MO). LC–MS/MS reagents were from Honeywell Burdick & Jackson® (Muskegon, MI), Fisher Scientific (Pittsburgh, PA) and Sigma-Aldrich (St. Louis, MO).

**Reaction knockout selection**. iJO1366[89] was used as the metabolic model for *E. coli* metabolism; GLPK (version 4.57) was used as the linear program solver. MCMC sampling[90] was used to predict the flux distribution of the optimized reference strain. Uptake, secretion, and growth rates were constrained to the measured average value ± SD. Potential reaction deletions were ranked by (1) averaged sampled flux, (2) the number of immediate upstream and downstream metabolites that could be measured, (3) the number of genes required to produce a functional enzyme. Reactions involved in sampling loops, that were spontaneous, were computationally or experimentally essential, or were not actively expressed under the experimental growth conditions were not included in the analysis. Also, reactions that would require more than one genetic alteration to abolish activity

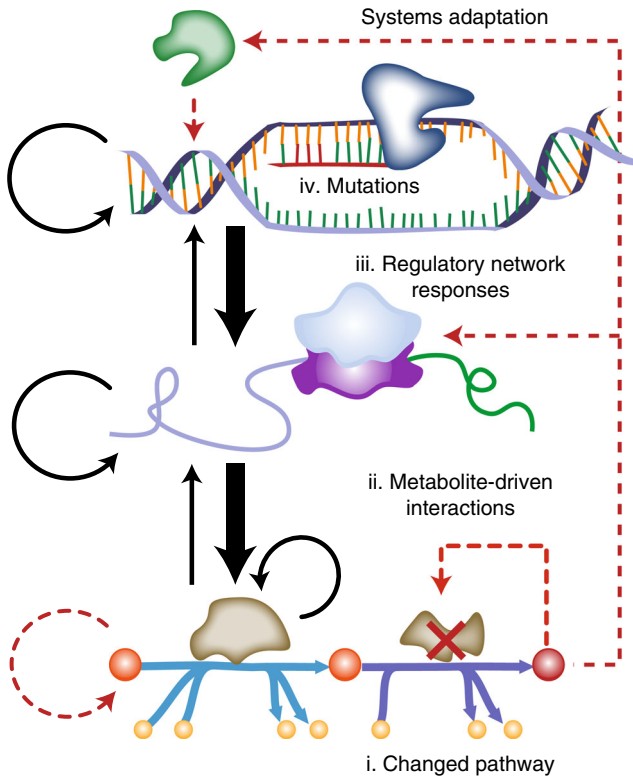

**Systems adaptation**

iv. Mutations

iii. Regulatory network responses

ii. Metabolite-driven interactions

i. Changed pathway usage

**Fig. 7** A model of biological systems adaptation following the KO of key metabolic enzymes. (i) Suboptimal pathway usage limited allocation of carbon to biomass precursors. (ii) Perturbed metabolite levels triggered transcription regulatory network (TRN) responses in the uKOs. (iii) Activation of the TRN revealed a hierarchy of regulation involving competing and overlapping regulatory interactions between various system components including DNA, RNA, and proteins. (iv) Mutations selected during adaptive evolution changed many regulatory networks, and also introduced innovations that targeted specific pathway or metabolite imbalances

were excluded. The top 9 reactions deletions from the rank ordered set of reactions that met the above criteria were chosen for implementation.

**Adaptive laboratory evolution (ALE)**. Cultures were serially propagated using a 100 μL passage volume in 15 mL working volume flasks. The cultures were grown in M9 minimal medium with 4 g/L glucose and kept at 37 °C and well-mixed for full aeration. Cultures were passed to fresh flasks during exponential growth and with nutrient excess once they had reached an $OD_{600}$ of 0.3 (Tecan Sunrise plate reader, equivalent to an $OD_{600}$ of ~1 on a traditional spectrophotometer with a 1 cm path length). Four $OD_{600}$ measurements were taken for each flask, and the relation between $ln(OD_{600})$ and time was used to calculate the culture growth rates.

**Phenomics**. Culture density were measured at 600 nm absorbance with a spectrophotometer and correlated to cell biomass. Substrate uptake and secretion rate samples were filtered through a 0.22 μm filter (PVDF, Millipore) and measured using refractive index (RI) detection by HPLC (Agilent 12600 Infinity) with a Bio-Rad Aminex HPX87-H ion exclusion column. The HPLC method was the following: injection volume of 10 μL and 5 mM H2SO4 mobile phase set to a flow rate and temperature of 0.5 mL/min and 45 °C, respectively.

**LC-MS/MS instrumentation and data processing**. Metabolites were acquired and quantified on an AB SCIEX Qtrap® 5500 mass spectrometer (AB SCIEX, Framingham, MA) and processed using MultiQuant® 3.0.1 as described previously[34]. Mass isotopomer distributions (MIDs) were acquired on the same instrument and processed using MultiQuant® 3.0.1 and PeakView® 2.2[35].

**Metabolomics**. Uniformly labeled *E. coli* cell extracts were used as internal standards[91]. The same batch of internal standards was used with all samples and calibrators. Two sets of calibration curves (before and after all samples) were used

to correlate peak height ratio to absolute concentration. Quality Control sample that were composed of all biological replicates were ran twice a day to check the consistence of quantitation. Solvent blanks were injected periodically to check for carryover. System suitability tests were injected at the start of each day to check instrument performance.

Metabolomics samples were acquired from triplicate cultures by sampling 1 mL of cell broth at an OD600 ~1.0[33]. Analytical blanks were made by pooling filtered medium that was re-sampled using the FSF filtration technique. All biological replicates and blanks were analyzed in duplicate. Unless otherwise noted, the intracellular values reported are derived from the average of the triplicates ($n = 6$). Metabolites in the analytical blanks that had a concentration greater than 80% of that found in the triplicate samples were not analyzed. Metabolites with a quantifiable variability (RSD ≥ 50%) in the quality control samples or any individual components with an RSD ≥ 80 were not used for analysis.

Missing values were imputed using Amelia II[92] (version 1.7.4, 1000 imputations). Remaining missing values were approximated as ½ the lower limit of quantification for the metabolite normalized to the biomass of the sample. Metabolite concentrations were log normalized to generate an approximately normal distribution using LMGene[93] (version 3.3, "mult" = "TRUE", "lowessnorm" = "FALSE") prior to statistical analysis. A Bonferroni-adjusted P-value cutoff of 0.01 as calculated from a Student's t-test was used to determine significance between metabolite concentration levels.

**Fluxomics**. Fluxomics samples were acquired from triplicate cultures (10 mL of cell broth at an OD600 ~ 1.0) using a modified version of the FSF technique as described previously[35]. MIDs were calculated from biological triplicates, each ran in analytical duplicates ($n = 6$). MIDs with an RSD greater than 50 were excluded. In addition, MIDs with a mass that was found to have a signal greater than 80% in unlabeled or blank samples were excluded. A previously validated genome-scale MFA model of *E. coli* with minimal alterations was used for all MFA estimations using INCA[94] (version 1.4) as described previously[36]. The model was constrained using MIDs as well as measured growth, uptake, and secretion rates. Best flux values that were used to calculate the 95% confidence intervals were estimated from 500 restarts.

The 95% confidence intervals were used as lower and upper bound reaction constraints for further constraint-based analyses. MFA derived constraints that violated optimality were discarded and re-sampled. The descriptive statistics (i.e., mean, median, interquartile ranges, min, max, etc.) for each reaction for each model were calculated from 5000 points sampled from 5000 steps using optGpSampler[95] (version 1.1), which resulted in an approximate mixed fraction of 0.5 for all models. A permuted P-value < 0.05 and geometric fold-change of sampled flux values > 0.001 were used to determine differential flux levels, differential metabolite utilization levels, and differential subsystem utilization levels between models. Demand reactions and reactions corresponding to unassigned, transport; outer membrane porin, transport; inner membrane, inorganic ion transport and metabolism, transport; outer membrane, nucleotide salvage pathway, oxidative phosphorylation were excluded from differential flux analysis. The geometric fold-change of the mean between models and the reference model were used for hierarchical clustering; the median, interquartile ranges, min, and max values of each sampling distribution for each reaction and model were used as representative samples for downstream statistical analyses.

**Transcriptomics**. Total RNA was sampled from triplicate cultures (3 mL of cell broth at an OD600 ~1.0) and immediately added to 2 volumes Qiagen RNA-protect Bacteria Reagent (6 mL), vortexed for 5 s, incubated at room temperature for 5 min, and immediately centrifuged for 10 min at 17,500 RPMs. The supernatant was decanted and the cell pellet was stored in the -80 °C. Cell pellets were then incubated with Readylyse Lysozyme, SuperaseIn, Protease K, and 20% SDS for 20 min at 37 °C. Total RNA was isolated and purified using the Qiagen RNeasy Mini Kit columns. On-column DNase-treatment was conducted for 30 minutes at 25 °C. RNA was quantified using a Nano drop and checked for quality using an RNA-nano chip on a bioanalyzer. The rRNA was removed using Epicentre's Ribo-Zero rRNA removal kit for Gram Negative Bacteria. A KAPA Stranded RNA-Seq Kit (Kapa Biosystems KK8401) was used following the manufacturer's protocol to create sequencing libraries with an average insert length of around ~300 bp for two of the three biological replicates. Libraries were ran on a MiSeq and/or HiSeq (Illumina).

RNA-Seq reads were aligned using Bowtie[96] (version 1.1.2 with default parameters). Expression levels for individual samples were quantified using Cufflinks[97] (version 2.2.1, library type fr-firststrand) Quality of the reads was assessed by tracking the percentage of unmapped reads and expression level of genes that mapped to the ribosomal gene loci rrsA-F and rrlA-F. All samples had a percentage of unmapped reads <7%. Differential expression levels for each condition ($n = 2$ per condition) compared to either the starting strain or initial knockout strain were calculated using Cuffdiff[97] (version 2.2.1, library type fr-firststrand, library norm geometric). Genes with an 0.05 FDR-adjusted P-value <0.01 were considered differentially expressed. Expression levels for individual samples for all combinations of conditions tested in downstream statistical analyses were normalized using Cuffnorm[97] (version 2.2.1, library type fr-firststrand, library norm geometric). Genes with unmapped reads were imputed using a bootstrapping

approach as coded in the R package Amelia II (version 1.7.4, 1000 imputations). Remaining missing values were filled using the minimum expression level of the data set. Normalized FPKM values for gene expression were log2 normalized to generate an approximately normal distribution prior to any statistical analysis. All replicates for a given condition were found to have a pairwise Pearson correlation coefficient of 0.95 or greater.

**DNA resequencing**. Total DNA was sample from an overnight culture (1 mL of cell broth at an OD600 of ~2.0) and immediately centrifuged for 5 min at 8000 RPMs. The supernatant was decanted and the cell pellet was frozen in the −80 °C. Genomic DNA was isolated using a Nucleospin Tissue kit (Macherey Nagel 740952.50) following the manufacturer's protocol, including treatment with RNase A. Resequencing libraries were prepared using a Nextera XT kit (Illumina FC-131-1024) following the manufacturer's protocol. Libraries were ran on a MiSeq (Illumina).

DNA resequencing reads were aligned to the *E. coli* reference genome (U00096.2, genbank) using Breseq[98] (version 0.26.0) as populations. Mutations with a frequency of <0.1, *P*-value >0.01, or quality score <6.0 were removed from the analysis. In addition, genes corresponding to *crl*, insertion elements (i.e, *insH1*, *insB1*, and *insA*), and the *rhs* and *rsx* gene loci were not considered for analysis due to repetitive regions that appear to cause frequent miscalls when using Breseq. mRNA and peptide sequence changes were predicted using BioPython (https://github.com/biopython/biopython.github.io/). Large regions of DNA (minimum of 200 consecutive indices) where the coverage was two times greater than the average coverage of the sample were considered duplications.

**Structural analysis**. Corresponding PDB files for genes with a mutation of interested were downloaded from PDB[99, 100]. Structural models for genes for which there were no corresponding PDB files were taken from I-TASSER generated homology models[101] or generated using the I-TASSER protocol[102]. The BioPython predicted sequence changes and important protein features as listed in EcoCyc[103] were visualized and annotated using VMD[104].

**System component statistical feature identification analyses**. Network components (i.e., RNA-seq, metabolomics, fluxomics, genomics) were pre-processed as described above, and subjected to a feature identification analysis pipeline. Network components for each lineage were first subjected to a differential test (ref vs. KO, KO vs. endpoints, ref vs. endpoints, and endpoints vs. endpoints). The criteria for significance for each of the data types are detailed below. Metabolomics: *P*-value < 0.01 and 0.5 < fold_change < 2.0 as calculated from a *t*-test of the g-log normalized metabolite concentrations. Transcriptomics: *q*-value (0.05 FDR corrected *P*-value) and abs (log2(fold-change)) > 1.0 as calculated by Cuffdiff. Fluxomics: *P*-value < 0.01 and abs (geometric fold_change) > 0.001 as calculated from re-sampled flux distributions that were constrained by the 95% confidence intervals derived from estimated MFA flux bounds (demand reactions and reactions in subsystems corresponding to unassigned, transport; outer membrane porin, transport; inner membrane, inorganic ion transport and metabolism, transport; outer membrane, nucleotide salvage pathway, oxidative phosphorylation were excluded). Mutations: frequency > 0.1 (mutations in the reference strain and in repetitive regions were excluded). Components that met the significance criteria for any combination of comparisons from the differential test were used in the pairwise PLS-DA analyses and profile matching. Counts of significant components for each lineage were based on components that met the significance criteria for Ref vs. eRef, or uKO vs. eKO.

Network components for each lineage were subjected to pairwise PLS-DA analyses (ref vs. KO, KO vs. endpoints, ref vs. endpoints, and endpoints vs. endpoints). The components with a loadings 1 magnitude within the top 25% of all components and correlation coefficient > 0.88 for different combinations of comparison were selected using pairwise PLS-DA analysis.

Network components for each lineage were subjected to profile matching. System component levels between Ref, eKO, and uKO were correlated (Pearson's R) to six profiles in both positive and negative directions. *novel−*, *novel+*, *overcompensation−*, *overcompensation+*, *partially restored−*, *partially restored+*, *reinforced−*, *reinforced+*, *restored−*, *restored+*, *unrestored−*, *unrestored +* profiles were encoded in integer form as 1-1-0, 0-0-1, 1-0-2, 1-2-0, 2-0-1, 0-2-1, 2-1-0, 0-1-2, 1-0-1, 0-1-0, 1-0-0, and 0-1-1. System components were binned into profiles when a Pearson correlation coefficient > 0.88 was calculated. Only negligible changes in the assignment of profiles were found when using absolute or relative component units (e.g., mmol*gDCW$^{-1}$ vs. log2(FC vs. ref)) or different correlation methods (i.e., Spearman).

**System component statistical sample trend analysis**. Components identified from the differential tests (except for metabolomics) were used for sample trend analyses. Hierarchical clustering was used to diagnose sample groupings and distances between samples (distance metric of Euclidean and linkage method of complete). Principal component analysis (PCA) as encoded in the R package pcaMethods[105] (version 1.64.0, univariate scaling, centering, SVD PCA) was then used as a representative unsupervised method to project samples into component space, and confirm the relative magnitude and direction of component weights. PCA models were first constructed for the reference, knockout, and endpoint for

each of the lineages to confirm that the primary component best separated the reference and endpoint from the knockout, and that the second component best separated the reference and knockout from the endpoint. PCA models were then constructed for the reference, knockout, and all endpoints for each network perturbation. The PCA models were validated using cross validation (CV type of Krzanowski, default 5 segment with 5 CV runs per segment with minimum number of segments equal to the number of samples). Partial Least Squares Discriminatory Analysis (PLS-DA) was implemented using the R package pls[106] (version 2.5, univariate scaling, centering, Canonical Powered Partial Least Squares (cppls) PLS-DA) was used to project replicate samples into component space. PLS-DA models were first constructed for the reference, knockout, and endpoint for each of the lineages to confirm that the primary component best separated the reference and endpoint from the knockout, and that the second component best separated the reference and knockout from the endpoint. PLS-DA models were then constructed for the reference, knockout, and all endpoints for each network perturbation. The PLS-DA models were validated using cross validation (default 10 segments with minimum number of segments equal to the number of samples).

The loadings distance (i.e., the difference in loadings values) between the ref and uKO strain along axis 1 (i.e., mode 1) was used as a threshold to determine whether an eKO strain matched the general mode 1 and mode 2 trends identified in section 2a. A relative distance for each eKO strain along axis 1 was calculated as follows: relative distance = distance(uKOj, eKOi,j)/distance(ref, uKOj) where i = endpoint replicate for a particular KO lineage and j = each KO lineage. An eKO strain with a relative distance greater than 70% along axis 1 was determined to match the trend.

**Metabolite, flux, and gene set enrichment analyses**. Metabolite and gene set enrichment analyses were conducted using the subsystem categories of iJO1366. Flux and metabolite flux sum set enrichment analyses were conducted using the subsystem categories of iDM2015. A *P*-value < 1e−3 (hypergeometric test) was used to test for enriched subsystems. Gene set enrichment analysis on differentially expressed genes was also performed using with R package topGO[107] with GO annotations for *E. coli*[108]. A *P*-value < 0.05 (Fischer statistic, parent–child algorithm[109]) was used to test for enriched biological processes and molecular functions.

**Network distance and graph analyses**. The inverse mean values from sampled flux distributions that were constrained by the 95% confidence intervals derived from estimated MFA flux bounds were used as weights in calculating the shortest path from metabolite A to B. The iDM2015 network was deconstructed into a directed acyclic graph with metabolites and reactions composing the nodes and the connections between metabolites and reactions composing the links. Metabolites that did not contain carbon were excluded from the graph network. In addition, metabolites corresponding to co2, co, mql8, mql8h2, 2dmmql8, 2dmmql8h2, q8, q8h2, thf, ACP were also excluded. Metabolites corresponding to udpglcur, adpglc, gam6p were substituted as glycogen_c, uacgam, uacgam, respectively, as they were not present in the lumped and reduced iDM2015 network. The A*star algorithm as implemented in the python package networkX (https://github.com/networkx/networkx) (version 1.11) was used to calculate the shortest path of the graph network. The distance from metabolite A to B was calculated as half minus 1 the computed shortest path.

A redistribution of flux was defined as a change in path or path length between the reference and knockout and endpoint or knockout and endpoint. A change in flux capacity was defined as a change in path or path length between the reference and knockout, but not between the knockout and endpoint.

Nodes (i.e., metabolites) were categorized as intermediates, carriers, biomass precursors, and/or nucleotide salvage products. The correlation (Spearman R, *P*-value < 0.05) between path and path length and metabolite level was calculated between intermediates and carriers, carriers and biomass precursors, intermediates and biomass precursors, carriers and nucleotide salvage products, and biomass precursors and nucleotide salvage products.

**Biomass to network component correlation analysis**. EcoCyc[103] subsystems for the following biomass producing pathways were used in the analysis: amines and polyamines biosynthesis, amino acids biosynthesis, nucleosides and nucleotides biosynthesis, fatty acid and lipid biosynthesis, cofactors, prosthetic groups, electron carriers biosynthesis, cell structures biosynthesis, and carbohydrates biosynthesis. Gene identifiers from these pathways were mapped onto iDM2014 via the GPR relation to identify biomass producing reactions and metabolites. The analysis was conducted at the level of individual lineages using the system component profiles of restored−, novel+, overcompensation−, partially restored−, and reinforced + to identify positively correlated (correlation coefficient > 0.88, Pearson, *r*) with growth (i.e., growth promoting) and negatively correlated (correlation coefficient < −0.88, Pearson, *r*) with growth (i.e., growth inhibiting). The number of significant biomass components were divided by the number of measured biomass components, and expressed as a percent. A direct pairwise correlation between metabolite concentrations, transcript levels, and fluxes, and growth rate was also performed (units of log2(FC vs. ref)) between the reference strain, knockout, and endpoints for all or each knockout condition for comparison (data not shown). Components that were

positively correlated (correlation coefficient > 0.88, Pearson, $r$) with growth rate or negatively correlated (correlation coefficient > 0.88, Pearson, $r$) with growth rate were identified.

**Inter- and intra-component correlation analysis**. A global pairwise correlation between metabolite concentrations, transcript levels, and fluxes was performed by comparing the agreement and disagreement between component profiles of restored+, novel+, overcompensation+, partially restored+, unrestored+, and reinforced+. Components with matching profiles with correlation coefficients > 0.88 (Pearson, $R$) were correlated; components with matching profiles with correlation coefficients < −0.88 (Pearson, $R$) were anti-correlated. A similar global pairwise correlation between metabolite concentrations, transcript levels, and fluxes was performed (units of log2(FC vs. ref)) for comparison (data not shown). Components with a correlation coefficient > 0.88 (Spearman, $r$) were correlated; Components with a correlation coefficient < −0.88 (Spearman, $r$) were anti-correlated.

**Regulation to network component correlation analysis**. Significantly correlated components were compared to annotated gene-to-reaction, and metabolite-to-reaction interactions annotations in iJO1366, and to annotated transcription factor-to-gene, metabolite-to-transcription factor, metabolite-to-transcription factor-to-gene, metabolite-to-transcript, and metabolite-to-reaction regulatory interactions from the EcoCyc database[103]. EcoCyc database identifier were mapped to iJO1366 identifiers using a combination of ChEBI[110], MetaNetX[111–113], EC numbers, InChi strings, and manual curation. The mode of component interactions were encoded as either positive for reactant-reaction, activating, or stabilizing interactions, or negative for product-reaction, inhibiting, or de-stabilizing interactions. The sign and magnitude of the correlation coefficient (Pearson, $r$) of matching categories was compared to the mode of interaction to determine agreement (correlation coefficient > 0.88 and positive mode, or correlation coefficient < −0.88 and negative mode). The inverse was used to determine disagreement.

The classification of global regulators follows the definition given by Martinez-Antonio et al.[46] Global transcription factors are defined to include CRP, IHF, FNR, FIS, ArcA, Lrp, and Hns. A secondary level of regulators are defined to include NarL, Fur, Mlc, CspA, Rob, PurR, PhoB, CpxR, and SoxR. The secondary level and lower level regulators (e.g., local transcription factors) were further broken into classes for local and general stresses.

**Regulator activation categorization**. A profile for the activation status of each regulator for each knockout evolution was determined. The analysis was first limited to regulated entities that had only a single annotated regulator. The analysis was then expanded to include all regulators and regulated entities. A category weight for each regulated entity for each endpoint was calculated as follows: weight, $i,j = \mathrm{abs}(\mathrm{corr},i,j)*1/(nEPs,i)*1/(nRegulators,k)$ where $i$ = endpoint, $j$ = category, $k$ = regulators, nEPs = number of endpoints per knockout evolution, corr = correlation coefficient, nRegulators = number of regulators per regulated entity. A confidence score for each regulator for each knockout was calculated as follows: confidence,$i = \mathrm{sum}(\mathrm{weight},i,j,k)$ where $i$ = knockout, $j$ = endpoint, and $k$ = regulated gene. A higher confidence score indicates a consistently higher correlation to the category across all regulated entities that are regulated by the regulator.

**Code availability**. Published software used in this study are noted in the Methods. Custom software used for the analyses presented in this study are deposited on Github (https://github.com/dmccloskey).

## Data availability

The data that support the findings of this study are available from the corresponding author upon reasonable request.

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

## Acknowledgements

We thank José Utrilla for helpful discussion and guidance when implementing the knockouts in the pre-evolved strain. We thank Jamey Young for helpful discussions throughout the MFA analysis. We thank Laurence Yang for helpful discussions regarding optimization and statistical analysis. This work was funded by the Novo Nordisk Foundation Grant Number NNF10CC1016517.

## Author contributions

D.M. designed the experiments; generated the strains; conducted all aspects of the metabolomics, fluxomics, phenomics, transcriptomics, and genomics experiments; performed all multi-omics statistical, graph, and modeling analyses; and wrote the manuscript. T.E.S. ran the ALE experiments. E.B. assisted with structural analysis. R.S. processed the DNA and RNA samples. S.X. assisted with metabolomics and fluxomics data collection, sample processing, and peak integration. Y.H. assisted with fluxomics data collection and sample processing. A.M.F designed and supervised the evolution experiments, and contributed to the data analysis and the manuscript. B.O.P conceived and outlined the study, supervised the data analysis, and co-wrote the manuscript.

## Additional information

**Competing interests:** The authors declare no competing interests.

