## [Peer Review File · Nature Communications]

Reviewers' comments:

Reviewer #1 (Remarks to the Author):

The manuscript of McCloskey addresses the question of how a bacterium responds to gene loss and how this initial physiological response is altered following compensatory adaptation in the lab. Based on collecting various layers of omics data (genome, transcriptome, metabolome, fluxome), the authors propose several mechanisms by which the cell copes with gene loss on physiological and evolutionary time scales. The manuscript also attempts to uncover some general principles underlying compensatory changes upon gene loss: i) the initial response is sub-optimal and become re-optimized during evolution, ii) the initial response often starts with local metabolite level perturbations that evoke a regulatory response, iii) many initial changes are restored during compensatory evolution, iv) still, parallel evolving lines tend to achieve unique solutions. All these conclusions are important and the data generated in this study is truly impressive. However, the supports for most of the conclusions are not clearly presented. In general, whether a particular claim is based on verbal speculations, rigorous computational inferences (e.g. based on systems modelling) or direct experimental manipulation is often blurred in the results section. I was left with the impression that some of the conclusions were based on rather indirect evidences only (see my specific concerns below). This isn't necessarily a problem as the study proposes a large number of biochemical mechanisms and some of them are more hypothetical than others. Yet, I would suggest to primarily focus on the strongest claims only, present them in a more convincing manner and clearly discuss the assumptions of those conclusions for which only indirect evidence exist. In general, the detailed arguments and type of evidence leading to each claim should be presented.

Major points:

1. Fitness compensation: A widely accepted definition of compensatory adaptation is a fitness increase during genetic adaptation which is disproportionately large in lines carrying deleterious mutations relative to that in evolving control lines (see PMID 10737410). From this perspective, the present study has two limitations. First, although the authors took great care to start from a reference line that is well adapted to the growth medium and also run control evolving lines, disproportionately large fitness increases were not formally demonstrated (i.e. by statistical tests). Based on Fig 1D, this could be easily shown as the reference (control) evolving lines did not show any appreciable fitness increase. Second, and more importantly, out of the 5 knockouts investigated, 2 did not show a significant fitness increase following laboratory evolution (based on final fitness, Supplementary Text). Although I can understand that this might be a consequence of the small initial fitness drop in these knockouts, it is still problematic because a formal evidence for compensatory adaptation is lacking. This makes it rather difficult to interpret the rest of the findings for these 2 knockouts and corresponding evolved lines. I suggest either to measure fitness with a more sensitive / less noisy method (direct competition, large numbers of replicates?) or to use data from the full evolutionary trajectories to demonstrate a fitness increase. For instance, a visual inspection of the trajectories of *sdhCB* indicates that a fitness increase did actually happen.

2. Definition of growth-limiting and growth inhibiting changes: According to Supplementary

Text, system components that increased / decreased in concentration / flux during evolution were considered as growth-limiting / growth inhibiting (btw, this definition should appear in the main text as well). While the categorization itself is useful, I'm not convinced that many of these cases represent genuine growth limitations or inhibitions. For example, observing a metabolite going up during evolution doesn't imply that it was growth limiting in the unevolved state. It could well be the case that the change in metabolite level is a correlated neutral, or even detrimental, response and not a causal one. A similar criticism applies to transcriptional changes. Flux changes might be easier to rationalize within this framework because one can use an FBA model to map between flux perturbations and growth rate. However, no such simple mappings can be reliably done for metabolite or gene expression levels. Thus, it would be more acceptable to present these system component changes in a descriptive manner, offer potential hypotheses to interpret them and validate some of the growth limiting / inhibiting changes by experimental manipulation. For example, the authors suggest many growth-inhibiting gene expression changes in the knockouts. If so, then experimentally downregulating specific genes should rescue some of the fitness defect.

3. Perturbed metabolite levels triggered transcription regulatory responses: Here the authors report many interesting findings by changed TF activations in the knockouts. However, it was unclear to me whether the 'strong evidence' for changed TF activation also incorporates information on the gene expression changes in the regulon of the TF. Specifically, I'd expect that if a TF activity is altered then genes belonging to its regulon should be enriched in up or downregulations. By any means, the type of evidence used for the inference should be made more transparent and briefly discussed in the main text.

4. It was not clear to me whether there is any direct evidence that misallocation of resources (i.e. upregulation of unproductive pathways) contributes to the fitness reduction in the uKOs. Such misallocations may have only a tiny fitness cost which might not be even detectable through standard growth rate assays. If there's no direct evidence in support of this scenario, I would suggest to present it as a hypothesis / open possibility.

5. Section v of the Results (pages 8 – 9) was difficult to follow in the main text. I had to go to the Supplementary text to better understand what sort of discrepancies the authors were referring to. One important discrepancy identified here was the lack of congruence between flux changes and gene expression changes. Importantly, this phenomenon has been reported before (e.g. PMID 27789812, 17898166) and was explained by a strong role of metabolite-level regulation (i.e. substrate / product / allosteric regulator concentrations are often the primary determinants of flux). This should be explicitly discussed. Furthermore, it should be more clearly described whether a particular discrepancy existed in the uKO strains or emerged in the eKO strains only.

6. Restoration of wild-type (reference) omics states: The authors argue that compensation tend to restore wild-type molecular states. The main observation supporting this notion comes from PLS-DA analysis (Fig 2). However, in many cases, the second mode of PLS-DA explains a similar amount of variance as the first mode. This indicates that differences between evolved knockouts and reference are comparable to differences between unevolved and evolved knockouts. It would be informative to directly compare the distances (in a reduced dimensional space that captures most of the variation) between the reference, unevolved and evolved knockouts to quantify the extent of restoration.

7. Metabolomics measurements: The authors applied an RSD \geq 80% cutoff to exclude metabolites that are highly variable across replicate measurements. This sounds unusually

lenient. Would a more stringent cutoff yield similar biological conclusions?

8. Introduction: numerous papers on experimental evolution are cited, but only few on compensatory evolution. Here're some more directly relevant citations, including studies on the compensation of gene loss: PMID 10737410, 25157590, 24516157, 19041751, 25722415

9. Discussion: The general principles outlined in Fig 7B should be discussed in the text as well. Furthermore, the connection between these inferred principles and findings from previous compensatory evolution studies should also be discussed, especially those that generated omics data (e.g. ref. #9, PMID 25157590).

Minor points:

Several citations are missing:

- description of figure S2: "See Table S[] for a breakdown of each biomass-producing pathway"
 - SM page 14: "Biomass Precursors, and/or Nucleotide Salvage Products as defined in Table S[]."
 - Data and Software Availability section of the SM: some accession codes are missing.
- Fig S2A-C: Percentage on the y-axes shouldn't take negative values

Reviewer #2 (Remarks to the Author):

In this very interesting paper, the authors set up an adaptive evolution experiment where strains with specific knock out mutations are adapted under the same conditions where they were optimized. The responses to the loss of the specific gene or pathway or then analyzed through a variety of omic methods, includes analysis of the changes in transcription patterns and metabolite levels. Overall, this very ambitious project is well described in the paper and the data, while complex, are presented just about as well as possible. While I have no overall major criticisms, I do have comments and queries, presented in their order of appearance in the manuscript.

1) One general comment. There must be some time between the creation of the knock out mutant strains and the initiation of the adaptive evolution experiment. This will undoubtedly include incubation in some medium other than that used in the ALE experiments....for example, some growth on LB agar plates and LB liquid medium, during the creation of the mutants. Is this be something that has to be considered at all when thinking about just when and where the genetic diversity began to appear that might have been selected for during the ALE experiment? Obviously, this is something that cannot be eliminated; it is a "cost of doing business" so to speak. But, it might be happening...that is, some adaptations might have started before the formal initiation of the experiment....should this at least be acknowledged in some way?

2) Pg. 4, starting line 69. Please provide some more description information for the genes or pathways being eliminated. In particular, the authors expect the reader to know exactly what the various components of the GLCptspp are. This section would benefit from a bit more narrative detail.

3) Line 83. Are the two "dominant modes" the "primary mode" and the "secondary" mode?

If so, then it sounds like the primary is more dominant than the secondary, so I don't know what dominant means. If primary really means "first" and secondary really means "second", in temporal order, then this should be explicitly stated. Further, line 94 refers to secondary "modes" in the plural. I don't think the reader can easily follow just what the authors mean. This is reading as jargon that might have significant meaning for the authors, but the reader needs to be brought along.

4) Further, the authors refer to the "optimized" state (line 86). A bit of narrative on just how this initial optimization was achieved which be beneficial to the reader. Again, it doesn't have to be long, but there is a significant back story here, and as currently presented, it is assumed that the reader has done their homework.

5) Line 129. Please define non-oxPPP. There are a few other times this term is used, but I'm not sure where it is defined.

6) Pg. 7, last paragraph. With respect to the observation that transcriptional networks are altered, the authors provide a possible explanation: that the concentrations of the metabolic activators for specific transcription factors change. They discount a second model, that expression patterns for the transcription factors change because they have measured global transcription patterns. However, there is a third possibility that is not addressed: it is possible that the protein stability of the transcription factors has changed and this would not be reflected by the measured transcript levels. There is precedence for the half-life of a protein varying greatly as a function of the physiology of the bacterial cell, changing in a non-linear relationship with mRNA transcript abundance. This concern should be addressed...or at least mentioned.

7) Line 155 and elsewhere throughout the manuscript. The authors need to be consistent with their nomenclature. Here, does *arcA* and *purr* refer to the genes or the gene products? As currently presented I cannot be sure because the gene references begin with a lower-case letter, but are not italicized. This needs to be standardized throughout the paper and in the figures. See also, line 166 for *uhpAB*, etc....

8) Line 158. The section sub-heading is redundant to the first sentence of the paragraph below.

9) With respect to the 5th mechanism, just how quickly are these responses abrogated? Do we know how many cycles of adaptation occur before these processes begin to ameliorate? It is not essential for the paper to do anything but show the endpoint results. But, one wonders just what the temporal scale is. How quickly can the cells truly adapt. You begin to get a flavor of this in Fig. 1D, but as currently presented, we do not know which of the seven identified processes are occurring at any given time.

10) Line 187. Please define "re-wire." It reads as jargon and can have many meanings to the reader. More specificity is needed.

11) Line 206. Similarly, please define "tuned". Here with respect to the MG pathway "in *tpi*". Not clear just what this means.

12) Line 215. Pyruvate probably doesn't need capitalization.

13) In Figure legends and figures, as mentioned above, please standardize all genotype and phenotype nomenclature in legends and figures. Gene names should start with lower-case and be italicized. Protein or gene product names should start with capital letters and not be italicized. As currently presented, it is confusing to those familiar with standard bacterial nomenclature.

14) Fig. 1A. What is the meaning of the blue and beige "clouds" around the cells? What is

the significance of the large vs. small red Xs in the cells?

15) Fig. 1B. There is no label on the Y-axis. Is it growth rate as in Fig. 1D?

16) Fig. 7A. I think this model would read better if it was presented top down, instead of bottom up as currently presented.

Reviewer #3 (Remarks to the Author):

This article by McCloskey and colleagues describes the generation and analysis of a large dataset pertaining to *E. coli* gene deletion knockout strains. The authors evolved these strains in the laboratory, and characterized changes in their genotypes and phenotypes after the evolutionary adaptation period. Throughout this adaptation, the strains had a chance to reach a growth rate closer to the one of the unperturbed organism. The goal of this study was to dissect changes at multiple levels (intracellular metabolites, metabolic fluxes, gene expression, genetic mutations), understand mechanistically how these changes restore growth, and identify general patterns across different mutant strains.

Unfortunately, while the data collected could in principle constitute a very rich resource for biological inference (and I appreciate the huge effort that must have been invested in collecting and processing the data), I found the arguments for both specific mechanistic/causal explanations and general principles extremely weak. Moreover, the broad question of how microbial cells respond to gene deletions is a very interesting one, but definitely not a novel one. Again, the amount and diversity of data collected for this work is impressive. But it is not clear to me that the data, in the way it is processed and presented, leads to any significant advancement in our general understanding of how bacteria cope with gene deletions. On top of this, I regret to say that the article was overall poorly written, lacking depth in the motivation, background and biological context, and making strong, often unjustified statements that use vague definitions and unclear language. Part of the problem, in my opinion, is that the authors tried to pack in this paper a large amount of data and analyses, making it close to impossible to really make convincing arguments for any individual sub-topic.

I will not be able to include in my assessment all the details of all the issues I found in the text, because there are too many, but I will focus on a few broad ones, and on some examples, that I hope the authors will find useful for future submissions.

1. The introduction is extremely short, superficial and uninformative. Given that lines 46-57 at page 3 are a synopsis of the work, the actual introduction amounts to 8 lines of text, four of which basically just mention that people study function through knockouts. Except for a list of papers (REFs 9-17) that dealt with adaptation to gene loss (and that the authors dismiss simply by saying that compensatory mutations have been poorly characterized), there is no justification for why a new study is needed. Other authors (several quoted in the above list), have characterized in fine details the adaptation of individual mutants. Even more surprisingly, the manuscript doesn't refer prior pioneering work (e.g. Fong et al., JBC 2006) that some of the authors themselves had published using some of the same knockout strains and a similar philosophy. Furthermore, even if modeling is invoked to justify the

choice of the mutants, there is no mention of several papers that dealt with predicting the immediate and adaptive response to gene loss. Overall, I must admit that even just after reading the introduction I was left with a big open question mark: why is this work important? What was missing before? What should I expect to learn?

2. I understand that there is a lot of material, and that the authors were trying to be brief. At the same time, I think that certain basic concepts should be very clear even without looking at the supplementary material. Several sentences were just incomprehensible to me, despite reading them a few times. For example, page 8, line 160: "TF responses resulted in a misallocation of resources or amplification of processes reducing fitness". The explanation of data collected was described in a total of two lines (81-82). What classes of metabolites were measured, and how? What fluxes? How was expression measured? Were the strains resequenced to identify mutations? Only at the end of the experiment? The Supplementary Material reports some of this information, but not in enough detail. Furthermore what presented in the main text is in my mind below the threshold of what would make the article readable. Same for the computational approach: "Decomposition methods revealed...". What data was the method applied to and why? All seems very vague and superficial. The description in the Supplementary Methods file contain some details but it is still poorly explained, and is very far from the kind of rigorous description that would enable another researcher to recapitulate the results.

3. Another overall big criticism I have is that the authors seem to draw a lot of mechanistic links between observations. However, I could not find clear justifications for most of these statements, in the sense that it is not clear what causes what. Most of the causal links the authors describe are, as far as I can tell, just unverified hypotheses. One example is the beginning of section iii (page 7): "it was found that perturbed metabolite levels triggered transcriptional regulatory network response". I could not find any justification for causality in the Supplementary Material or Methods. Causality is also not proven at another very important level: appearance of specific mutations can be strongly indicative of function, but it is in general not obvious to establish whether and how a specific mutation affects fitness. Other authors have gone to great lengths to prove these connections, e.g. by reinserting specific mutations on the background of the unevolved strain (individually and in combinations). Other approaches may be possible, but it is not clear to me that the authors of this paper can prove causality through their structural arguments. I don't know that such a painstaking process would be necessary for publishing this work, but then I would be much more careful about the statements being made.

4. Last, I was not convinced that the general principles outlined in the sections of the manuscript, and summarized in Fig. 7, are in fact general principles of broad interest. First of all, some of the principles seem to me fairly trivial and understood based on abundant evidence of degree of interactions between cellular components. Second, the nontrivial aspects (e.g. that primary drivers are metabolites) are based on inferences of causality that, as mentioned above, are not really proven, but just hypothetical.

Response to Reviewers

Table of Contents:

Table of Contents:	1
Author's comments:	2
Reviewers' comments:	3
Reviewer #1 (Remarks to the Author):	3
Reviewer #2 (Remarks to the Author):	7
Reviewer #3 (Remarks to the Author):	10

Author's comments:

We would like to thank each of the reviewers for the time, detail, and thoroughness that was taken in reading, dissecting, and analyzing the manuscript. Given the complex nature and ambition of the study, it cannot be emphasized enough our appreciation for your effort in providing a multitude of helpful and beneficial critiques and comments.

We have taken the time to synthesize each of the reviewers comments, which are summarized in the remainder of this paragraph. Reviewer one has identified several weakness in the support for the arguments and conclusions that were made. These are based primarily around clearly defining whether an argument and conclusion is based on a) speculation, b) computational inference, or c) direct experimental evidence. Reviewer two has identified a multitude of poorly defined terms and jargon that will be confusing and non-obvious to most readers. Reviewer two has also identified some interesting points and details that could be better expanded in the text. Reviewer three has identified several major concerns in regards to the content and structure of the manuscript. These concerns include a) weak evidence for identified specific mechanisms of adaptation as well as general principles of adaptation, b) questions regarding the overall novelty of investigating gene loss and additional contributions this manuscript makes to the topic, and c) problems arising from an effort to pack in too much information into too short of a format.

Based on the reviewer comments, we have extensively revised the manuscript. This has entailed splitting the manuscript into different contributions. The contribution that we are re-submitting here is focused on the high level commonalities and lessons learned from detailed -omics analysis and bioinformatics. Contributions dedicated to the systems biology of each KO ALE experiment will be submitted separately. In our previous effort, due to the fact that we were trying to pack in too much information into too short of a format (as described by all reviewers) novelties uncovered in the omics data were lost. Even worse, important support for arguments and conclusions were non obvious because much of the support was relegated to the supplemental material. By splitting up the manuscript, it has allowed us to more clearly describe the experimental and informatics methods used, it has allowed us better link the arguments and claims made with the experimental evidence, and it has allowed us to expand upon novel results uncovered in the omics data.

Our detailed response to each of the specific comments of each of the reviewers are given in the sections below.

Reviewers' comments:

Reviewer #1 (Remarks to the Author):

The manuscript of McCloskey addresses the question of how a bacterium responds to gene loss and how this initial physiological response is altered following compensatory adaptation in the lab. Based on collecting various layers of omics data (genome, transcriptome, metabolome, fluxome), the authors propose several mechanisms by which the cell copes with gene loss on physiological and evolutionary time scales. The manuscript also attempts to uncover some general principles underlying compensatory changes upon gene loss: i) the initial response is sub-optimal and become re-optimized during evolution, ii) the initial response often starts with local metabolite level perturbations that evoke a regulatory response, iii) many initial changes are restored during compensatory evolution, iv) still, parallel evolving lines tend to achieve unique solutions. All these conclusions are important and the data generated in this study is truly impressive. However, the supports for most of the conclusions are not clearly presented. In general, whether a particular claim is based on verbal speculations, rigorous computational inferences (e.g. based on systems modelling) or direct experimental manipulation is often blurred in the results section. I was left with the impression that some of the conclusions were based on rather indirect evidences only (see my specific concerns below). This isn't necessarily a problem as the study proposes a large number of biochemical mechanisms and some of them are more hypothetical than others. Yet, I would suggest to primarily focus on the strongest claims only, present them in a more convincing manner and clearly discuss the assumptions of those conclusions for which only indirect evidence exist. In general, the detailed arguments and type of evidence leading to each claim should be presented.

Major points:

1. Fitness compensation: A widely accepted definition of compensatory adaptation is a fitness increase during genetic adaptation which is disproportionately large in lines carrying deleterious mutations relative to that in evolving control lines (see PMID 10737410). From this perspective, the present study has two limitations. First, although the authors took great care to start from a reference line that is well adapted to the growth medium and also run control evolving lines, disproportionately large fitness increases were not formally demonstrated (i.e. by statistical tests). Based on Fig 1D, this could be easily shown as the reference (control) evolving lines did not show any appreciable fitness increase. Second, and more importantly, out of the 5 knockouts investigated, 2 did not show a significant fitness increase following laboratory evolution (based on final fitness, Supplementary Text). Although I can understand that this might be a consequence of the small initial fitness drop in these knockouts, it is still problematic because a formal evidence for compensatory adaptation is lacking. This makes it rather difficult to interpret the rest of the findings for these 2 knockouts and corresponding evolved lines. I suggest either to measure fitness with a more sensitive / less noisy method (direct competition, large numbers of replicates?) or to use data from the full evolutionary trajectories to

demonstrate a fitness increase. For instance, a visual inspection of the trajectories of *sdhCB* indicates that a fitness increase did actually happen.

We would like to thank the reviewer for bringing up the important topic of fitness compensation and how it pertains to this manuscript and ALE in general. The reviewer has highlighted two main concerns: 1) A non significant change in fitness of the control lineages and 2) a non significant change in fitness of the *u/eGND* and *u/eSdhCB* lineages.

First, the non significant change in fitness of the control lineages was expected. The control lineages were included to demonstrate that our experimental design (i.e., starting from a pre-evolved strain) minimized any confounding variables brought about by adaption to the growth conditions of the experiment. As shown in Figure 1E, only minimal changes in -omics data were found in the two control lines. This is a major differentiator of our work from prior work, and has allowed us to clearly differentiate the changes in -omics data brought about by the individual knockouts as shown in Figure 2. We do not think that the success of the control lineages in validating our unique experiment design should be counted against us.

Second, while we did not find any significant changes in fitness (determined from growth rate as measured under batch cultivation) *u/eGND* and *u/eSdhCB* lineages, we did find massive changes in all -omics data measured. This is an interesting finding that further demonstrates the robustness of biochemical networks to perturbations (i.e., that massive changes in metabolic and regulatory network can occur while only minimal changes in fitness occur). It is also interesting that the -omics data in these strains follows the same trends (Figure 2 and 3) as the other strains where large changes in fitness were found. This observations highlights an important question as to whether fitness (as determined by growth rate or even a competitive assay) is the best criteria for determining whether compensatory adaptation can occur. From the findings of this study, we would postulate that perhaps a more precise criteria for determining a shift from and to a fitness plateau would be the underlying -omics data (as shown in Figure 2). While beyond the scope of this study, if further experiments are conducted where massive changes in omics data are found during evolution while only minimal changes in fitness occur, then perhaps the field may consider to build a new consensus definition for the term 'compensatory adaptation.'

2. Definition of growth-limiting and growth inhibiting changes: According to Supplementary Text, system components that increased / decreased in concentration / flux during evolution were considered as growth-limiting / growth inhibiting (btw, this definition should appear in the main text as well). While the categorization itself is useful, I'm not convinced that many of these cases represent genuine growth limitations or inhibitions. For example, observing a metabolite going up during evolution doesn't imply that it was growth limiting in the unevolved state. It could well be the case that the change in metabolite level is a correlated neutral, or even detrimental, response and not a causal one. A similar criticism applies to transcriptional changes. Flux changes might be easier to rationalize within this framework because one can use an FBA model to map between flux perturbations and growth rate. However, no such simple mappings can be reliably done for metabolite or gene expression levels. Thus, it would be more

acceptable to present these system component changes in a descriptive manner, offer potential hypotheses to interpret them and validate some of the growth limiting / inhibiting changes by experimental manipulation. For example, the authors suggest many growth-inhibiting gene expression changes in the knockouts. If so, then experimentally downregulating specific genes should rescue some of the fitness defect.

The use of growth-limiting and growth-inhibiting were meant to categorize changes that were found at the various -omics levels in relation to a reduction in fitness. It should be emphasized that these definitions were meant to be purely descriptive, and provide a way of grouping metabolites, fluxes, and gene expression levels that were suboptimally lowered or suboptimally elevated following gene knockout. Given the problematic nature that the reviewers have highlighted in regards to these definitions being correlative without evidence for causality, this section has been removed from the manuscript.

3. Perturbed metabolite levels triggered transcription regulatory responses: Here the authors report many interesting findings by changed TF activations in the knockouts. However, it was unclear to me whether the 'strong evidence' for changed TF activation also incorporates information on the gene expression changes in the regulon of the TF. Specifically, I'd expect that if a TF activity is altered then genes belonging to its regulon should be enriched in up or downregulations. By any means, the type of evidence used for the inference should be made more transparent and briefly discussed in the main text.

We apologize for the poor description of the method used to identify changed TF activity. The method is based on changes in the metabolite levels as well as changes in the regulons controlled by the TFs. We have dedicated a main text figure (Fig. 4) to better illustrate the method. We have included method details that were relegated to the supplemental material the the figure legend as well as the main text where appropriate. We have also included additional examples in the main text to increase the transparency of TF activation analysis.

4. It was not clear to me whether there is any direct evidence that misallocation of resources (i.e. upregulation of unproductive pathways) contributes to the fitness reduction in the uKOs. Such misallocations may have only a tiny fitness cost which might not be even detectable through standard growth rate assays. If there's no direct evidence in support of this scenario, I would suggest to present it as a hypothesis / open possibility.

The reviewer is absolutely correct that there is no direct evidence that a general misallocation of resources is sufficient to contribute to fitness loss. We have qualified our conclusions in regards to the misallocation of resources as it pertained to gene expression to indicate the hypothetical nature of the discussion. Instead, we focused the discussion on the contribution of metabolites to the alterations seen in TF activity and gene expression.

5. Section v of the Results (pages 8 – 9) was difficult to follow in the main text. I had to go to the Supplementary text to better understand what sort of discrepancies the authors were referring to. One important discrepancy identified here was the lack of congruence between flux changes

and gene expression changes. Importantly, this phenomenon has been reported before (e.g. PMID 27789812, 17898166) and was explained by a strong role of metabolite-level regulation (i.e. substrate / product / allosteric regulator concentrations are often the primary determinants of flux). This should be explicitly discussed. Furthermore, it should be more clearly described whether a particular discrepancy existed in the uKO strains or emerged in the eKO strains only.

Based on the reviewers comments, and the confusion described by all reviewers, we have heavily revised this section. First, we have included the discussion that was relegated to the supplemental discussion to the main text, and have explicitly discussed the general findings in the context of the previous work mentioned by the reviewer. Second, we have dedicated a section called “Components profiles reveal systematic variations between ALE lineages, KOs, and measured data” that correspond to Figure 3 that better illustrate the analysis that was done to derive the component profiles that were used for comparison. In short, the comparison between layers of omics data was not based on differences between absolute values, but differences in the profile changes between the Reference strain, uKO, and eKO strains. Hence, we never state that a discrepancy is found in either the Reference strain, uKO, or eKO strains, but a discrepancy is found in the expected profile of related system components from the Reference time point, to the uKO time point, to the eKO time point.

6. Restoration of wild-type (reference) omics states: The authors argue that compensation tend to restore wild-type molecular states. The main observation supporting this notion comes from PLS-DA analysis (Fig 2). However, in many cases, the second mode of PLS-DA explains a similar amount of variance as the first mode. This indicates that differences between evolved knockouts and reference are comparable to differences between unevolved and evolved knockouts. It would be informative to directly compare the distances (in a reduced dimensional space that captures most of the variation) between the reference, unevolved and evolved knockouts to quantify the extent of restoration.

Was this done?

7. Metabolomics measurements: The authors applied an RSD \geq 80% cutoff to exclude metabolites that are highly variable across replicate measurements. This sounds unusually lenient. Would a more stringent cutoff yield similar biological conclusions?

For the analysis and mechanisms presented here, the cutoff yields no biological differences. Even without the cutoff, high variant metabolites would be filtered out by the significance criteria, which was applied before any other analysis was done.

8. Introduction: numerous papers on experimental evolution are cited, but only few on compensatory evolution. Here're some more directly relevant citations, including studies on the compensation of gene loss: PMID 10737410, 25157590, 24516157, 19041751, 25722415

Thank you for suggesting these references. We have now included the references in the introduction section.

9. Discussion: The general principles outlined in Fig 7B should be discussed in the text as well. Furthermore, the connection between these inferred principles and findings from previous compensatory evolution studies should also be discussed, especially those that generated omics data (e.g. ref. #9, PMID 25157590).

The reviewer is absolute correct. We have removed the discussion of general principles from the current contribution.

Minor points:

Several citations are missing:

- description of figure S2: "See Table S[] for a breakdown of each biomass-producing pathway"
 - SM page 14: "Biomass Precursors, and/or Nucleotide Salvage Products as defined in Table S[]."
 - Data and Software Availability section of the SM: some accession codes are missing.
- Fig S2A-C: Percentage on the y-axes shouldn't take negative values

Thank you for identifying these missing citations. The missing citations have been fixed.

Reviewer #2 (Remarks to the Author):

In this very interesting paper, the authors set up an adaptive evolution experiment where strains with specific knockout mutations are adapted under the same conditions where they were optimized. The responses to the loss of the specific gene or pathway or then analyzed through a variety of omic methods, includes analysis of the changes in transcription patterns and metabolite levels. Overall, this very ambitious project is well described in the paper and the data, while complex, are presented just about as well as possible. While I have no overall major criticisms, I do have comments and queries, presented in their order of appearance in the manuscript.

1) One general comment. There must be some time between the creation of the knock out mutant strains and the initiation of the adaptive evolution experiment. This will undoubtedly include incubation in some medium other than that used in the ALE experiments....for example, some growth on LB agar plates and LB liquid medium, during the creation of the mutants. Is this be something that has to be considered at all when thinking about just when and where the genetic diversity began to appear that might have been selected for during the ALE experiment? Obviously, this is something that cannot be eliminated; it is a "cost of doing business" so to speak. But, it might be happening...that is, some adaptations might have started before the formal initiation of the experiment....should this at least be acknowledged in some way?

The question of adaptation occurring before the formal initiation of the experiment is a valid concern that is worth mentioning. It is true that in order to generate the knockout mutants growth in LB and growth on agar plates is required. This cannot be avoided when following

current microbiology best practices. We have added to the methods that the starting ALE cultures were inoculated from frozen glycerol stocks in order to minimize the chance of compensatory mutations arising in any of the pre cultures prior to the formal initiation of the ALE experiment.

2) Pg. 4, starting line 69. Please provide some more description information for the genes or pathways being eliminated. In particular, the authors expect the reader to know exactly what the various components of the GLCptspp are. This section would benefit from a bit more narrative detail.

We have improved the description for the gene and pathways eliminated in this contribution.

3) Line 83. Are the two “dominant modes” the “primary mode” and the “secondary” mode? If so, then it sounds like the primary is more dominant than the secondary, so I don’t know what dominant means. If primary really means “first” and secondary really means “second”, in temporal order, then this should be explicitly stated. Further, line 94 refers to secondary “modes” in the plural. I don’t think the reader can easily follow just what the authors mean. This is reading as jargon that might have significant meaning for the authors, but the reader needs to be brought along.

Thank you for highlighting this confusing use of jargon. We have removed the use of the term “dominant modes” and have instead replaced it with the “first two modes”. The use of “modes” on the line 94 was a typo. This has been corrected.

4) Further, the authors refer to the “optimized” state (line 86). A bit of narrative on just how this initial optimization was achieved which be beneficial to the reader. Again, it doesn’t have to be long, but there is a significant back story here, and as currently presented, it is assumed that the reader has done their homework.

Thank you for identifying our poor explanation and definition of “optimized” state. We have clarified our use of the term in the main text. When appropriate, we have also substituted the word “optimized” to “evolved” to make it more clear for the reader.

5) Line 129. Please define non-oxPPP. There are a few other times this term is used, but I’m not sure where it is defined.

We have better defined this acronym in the main text.

6) Pg. 7, last paragraph. With respect to the observation that transcriptional networks are altered, the authors provide a possible explanation: that the concentrations of the metabolic activators for specific transcription factors change. They discount a second model, that expression patterns for the transcription factors change because they have measured global transcription patterns. However, there is a third possibility that is not addressed: it is possible that the protein stability of the transcription factors has changed and this would not be reflected

by the measured transcript levels. There is precedence for the half-life of a protein varying greatly as a function of the physiology of the bacterial cell, changing in a non-linear relationship with mRNA transcript abundance. This concern should be addressed...or at least mentioned.

This is a very good point that is brought up by the reviewer. In addition to protein stability, it could also be included the up or down regulation of proteases that may contribute to the half-life of the transcription factors. This is something that current technology does not allow us to measure well. We have therefore added this as a potential alternative explanation to the observed alterations in the transcription network.

7) Line 155 and elsewhere throughout the manuscript. The authors need to be consistent with their nomenclature. Here, does *arcA* and *purr* refer to the genes or the gene products? As currently presented I cannot be sure because the gene references begin with a lower-case letter, but are not italicized. This needs to be standardized throughout the paper and in the figures. See also, line 166 for *uhpAB*, etc....

The reviewer has highlighted an inconsistency that was missed in the writing and editing of the manuscript. We have standardized our italicization of genes and capitalization of gene products.

8) Line 158. The section sub-heading is redundant to the first sentence of the paragraph below.

Thank you for identifying this redundancy. We gone through each of the sub-headings and first paragraph sentences, and changed the word usage so as not be be redundant.

9) With respect to the 5th mechanism, just how quickly are these responses abrogated? Do we know how many cycles of adaptation occur before these processes begin to ameliorate? It is not essential for the paper to do anything but show the endpoint results. But, one wonders just what the temporal scale is. How quickly can the cells truly adapt. You begin to get a flavor of this in Fig. 1D, but as currently presented, we do not know which of the seven identified processes are occurring at any given time.

This is a very interesting question that the reviewers have identified on the dynamics of adaptive changes, and the order in which these changes occur. We feel that this question is beyond the scope of the current work. However, it can be noted that follow up studies are currently underway to address this question. It would be possible to discuss preliminary results, but we do feel that thorough analysis and discussion of those results would best be included in another contribution.

10) Line 187. Please define "re-wire." It reads as jargon and can have many meanings to the reader. More specificity is needed.

Thank you for identifying another piece of jargon. We have replaced the use of "re-wire" with "changed" or "altered".

11) Line 206. Similarly, please define “tuned”. Here with respect to the MG pathway “in tpi ”. Not clear just what this means.

Similar to the above, we have replaced the use of the word “tune” and specifically referenced “alterations” in the expression of the pathway..

12) Line 215. Pyruvate probably doesn’t need capitalization.

We have changed this following the reviewers suggestion.

13) In Figure legends and figures, as mentioned above, please standardize all genotype and phenotype nomenclature in legends and figures. Gene names should start with lower-case and be italicized. Protein or gene product names should start with capital letters and not be italicized. As currently presented, it is confusing to those familiar with standard bacterial nomenclature.

Similar to comment 7, we have corrected this inconsistency.

14) Fig. 1A. What is the meaning of the blue and beige “clouds” around the cells? What is the significance of the large vs. small red Xs in the cells?

The blue outlines are meant to signify optimized while the red outlines are meant to signify perturbed. The large X is meant to emphasize the impact of the initial knockout on fitness, while the small X is meant to emphasize the adaption to the initial knockout at the end of evolution.

15) Fig. 1B. There is no label on the Y-axis. Is it growth rate as in Fig. 1D?

Similar to Figure 1D, the y-axis of Figure 1B is growth rate.

16) Fig. 7A. I think this model would read better if it was presented top down, instead of bottom up as currently presented.

We chose to present the model top down as this is the traditional view that most readers have of biology.

Reviewer #3 (Remarks to the Author):

This article by McCloskey and colleagues describes the generation and analysis of a large dataset pertaining to *E. coli* gene deletion knockout strains. The authors evolved these strains in the laboratory, and characterized changes in their genotypes and phenotypes after the evolutionary adaptation period. Throughout this adaptation, the strains had a chance to reach a growth rate closer to the one of the unperturbed organism. The goal of this study was to dissect

changes at multiple levels (intracellular metabolites, metabolic fluxes, gene expression, genetic mutations), understand mechanistically how these changes restore growth, and identify general patterns across different mutant strains.

Unfortunately, while the data collected could in principle constitute a very rich resource for biological inference (and I appreciate the huge effort that must have been invested in collecting and processing the data), I found the arguments for both specific mechanistic/causal explanations and general principles extremely weak. Moreover, the broad question of how microbial cells respond to gene deletions is a very interesting one, but definitely not a novel one. Again, the amount and diversity of data collected for this work is impressive. But it is not clear to me that the data, in the way it is processed and presented, leads to any significant advancement in our general understanding of how bacteria cope with gene deletions. On top of this, I regret to say that the article was overall poorly written, lacking depth in the motivation, background and biological context, and making strong, often unjustified statements that use vague definitions and unclear language. Part of the problem, in my opinion, is that the authors tried to pack in this paper a large amount of data and analyses, making it close to impossible to really make convincing arguments for any individual sub-topic.

We would like to thank the reviewer for this critical, but fair, assessment. We have made major revisions to the text to clarify our novelty, improved our presentation of the data and analyses to better illuminate the insights that were found regarding bacterial adaptation to gene loss, and improve the quality of the overall manuscript. Key to these efforts was expanding upon the analysis and improving the focus of the manuscript so as not to not pack too much data and analysis into too short of a format. Instead, we decreased the number of subsections, and expanded the description of the analyses done, the results that were obtained, and selected examples used to illustrate the results found. We have revised all figures to better align with the analyses and results obtained in order to better convey the high level insights of bacterial adaptation to gene loss found. We have also included additional analyses that were not previously included to better support our arguments.

I will not be able to include in my assessment all the details of all the issues I found in the text, because there are too many, but I will focus on a few broad ones, and on some examples, that I hope the authors will find useful for future submissions.

1. The introduction is extremely short, superficial and uninformative. Given that lines 46-57 at page 3 are a synopsis of the work, the actual introduction amounts to 8 lines of text, four of which basically just mention that people study function through knockouts. Except for a list of papers (REFs 9-17) that dealt with adaptation to gene loss (and that the authors dismiss simply by saying that compensatory mutations have been poorly characterized), there is no justification for why a new study is needed. Other authors (several quoted in the above list), have characterized in fine details the adaptation of individual mutants. Even more surprisingly, the manuscript doesn't refer prior pioneering work (e.g. Fong et al., JBC 2006) that some of the authors themselves had published using some of the same knockout strains and a similar philosophy. Furthermore, even if modeling is invoked to justify the choice of the mutants, there

is no mention of several papers that dealt with predicting the immediate and adaptive response to gene loss. Overall, I must admit that even just after reading the introduction I was left with a big open question mark: why is this work important? What was missing before? What should I expect to learn?

We would like to thank the reviewer for providing a very actionable set of questions to guide our revision of the introduction. Following the suggestions of Reviewer 1 as well, we have expanded the introduction and have added additional citations for prior work. In particular, we have addressed concerns of the importance of the work, what was missing from previous work, and what the reader can expect to learn from this new addition to the field.

2. I understand that there is a lot of material, and that the authors were trying to be brief. At the same time, I think that certain basic concepts should be very clear even without looking at the supplementary material. Several sentences were just incomprehensible to me, despite reading them a few times. For example, page 8, line 160: “TF responses resulted in a misallocation of resources or amplification of processes reducing fitness”. The explanation of data collected was described in a total of two lines (81-82). What classes of metabolites were measured, and how? What fluxes? How was expression measured? Were the strains resequenced to identify mutations? Only at the end of the experiment? The Supp Material reports some of this information, but not in enough detail. Furthermore what presented in the main text is in my mind below the threshold of what would make the article readable. Same for the computational approach: “Decomposition methods revealed...”. What data was the method applied to and why? All seems very vague and superficial. The description in the Supplementary Methods file contain some details but it is still poorly explained, and is very far from the kind of rigorous description that would enable another researcher to recapitulate the results.

The reviewer is correct that in our effort to be brief, key pieces of information required for a full comprehension of the analysis, arguments, and conclusion were missing. We have included additional methodological details in the “Experimental Design” section of the results that answer the questions of from what strains were the data collected, what data was collected, and how the data was collected. We have also refrained from the use of vague statements as noted by the reviewer. For example, “Decomposition methods revealed” has been replaced by “Partial Least Squares Discriminatory Analysis revealed”.

3. Another overall big criticism I have is that the authors seem to draw a lot of mechanistic links between observations. However, I could not find clear justifications for most of these statements, in the sense that it is not clear what causes what. Most of the causal links the authors describe are, as far as I can tell, just unverified hypotheses. One example is the beginning of section iii (page 7): “it was found that perturbed metabolite levels triggered transcriptional regulatory network response”. I could not find any justification for causality in the Supplementary Material or Methods. Causality is also not proven at another very important level: appearance of specific mutations can be strongly indicative of function, but it is in general not obvious to establish whether and how a specific mutation affects fitness. Other authors have gone to great lengths to prove these connections, e.g. by reinserting specific mutations on the

background of the unevolved strain (individually and in combinations). Other approaches may be possible, but it is not clear to me that the authors of this paper can prove causality through their structural arguments. I don't know that such a painstaking process would be necessary for publishing this work, but then I would be much more careful about the statements being made.

The reviewers have highlighted an important point about what evidence constitutes causality. One advantage that this study has over previous studies in inferring causality is the breadth of omics data that we have to capture the interactions between a change in a component in one layer (e.g., metabolites) leads to changes in components in another layer (e.g., transcripts). For example, our mutation analysis of *galR* and *malT* is based on the interaction of the mutation and operons controlled by that TF that were found in the omics data. It is the consistency between omics data that makes these analyses more powerful.

In regards to the mutation analysis in general, the effect a given mutation has on a protein structure is inferred by bioinformatics (e.g., a mutation leading to a truncated peptide) or structural analysis (e.g., change in an amino acid residue known to interact with a substrate). We have omitted detailed discussion on these analyses in this contribution, and have instead dedicated individual contributions to each KO to dive further into each mutation, and the effect each contribution has on each level of omics data measured. In these separate contributions, we have also been more careful in how we present the fitness advantages of mutations identified. Specifically, we were careful to use terms like "may" or "could" to indicate our hypothesis in how a given mutation allowed for a fitness advantage. We also more clearly establish the background problem induced by the KO, and the solution that the mutation may present to overcome the biochemical challenge. For example, we first discussed the imbalance in redox carriers and changes in fluxes in the trans hydrogenases prior to discussing the three mutations that occurred in the trans hydrogenases.

4. Last, I was not convinced that the general principles outlined in the sections of the manuscript, and summarized in Fig. 7, are in fact general principles of broad interest. First of all, some of the principles seem to me fairly trivial and understood based on abundant evidence of degree of interactions between cellular components. Second, the nontrivial aspects (e.g. that primary drivers are metabolites) are based on inferences of causality that, as mentioned above, are not really proven, but just hypothetical.

The reviewer is correct for criticizing the general principles originally outline. We have removed this discussion from this contribution.

Reviewers' comments:

Reviewer #1 (Remarks to the Author):

I thank the authors for their significant efforts made towards improving the manuscript. The revised version is much more focused and improved a lot in clarity. The authors also addressed most of my previous concerns. However, few issues were not fully addressed and the revision process raised some new questions as well that should be addressed before publication.

1. Compensatory adaptation is widely defined as a fitness increase that is disproportionately greater in genetic backgrounds carrying a harmful mutation compared to a control. However, the authors didn't find a significant fitness increase in the evolved lines of two knockouts. Interestingly, even these lines show large changes in omics data. Based on these observations, the authors argue in their response letter that maybe it's time to revise the definition of compensatory adaptation. I'm not convinced by the latter argument and I doubt many geneticists / evolutionary biologists would be so.

In my eyes, this discrepancy either represents a lack of evidence for omics changes driven by compensatory mutations (could neutral mutations be involved?) or lack of resolution to measure small fitness increases. In either case, the degree of fitness and omics changes are largely decoupled suggesting that massive network rewiring can take place without any substantial fitness gain. Obviously, this raises the possibility that some of the omics changes seen in the other evolved lines, where fitness improved, might be irrelevant for compensation. For this reason, the issue of lack of fitness compensation should be openly discussed (mentioned in the results section and further interpreted in the conclusion).

2. Open issues and caveats are not discussed at all in the conclusion / discussion. This would be important as most conclusions drawn about molecular mechanisms are based on correlative evidences instead of experimental manipulations. Clearly, the large amount of omics data cannot substitute for experiments designed to prove specific hypotheses about causality. In this sense, there is a room for future works on demonstrating the causality of specific mutations, metabolite concentration changes, etc. This should be discussed upfront.

3. Intro, line 54: many of the references cited here (refs# 10-23) are not about compensatory evolution. I suggest to keep only the directly relevant citations to avoid confusion.

4. I couldn't retrieve the list of specific mutations in each genetic background. The readers will surely be interested in this piece of information.

5. Lines #163-164: here the authors report that lineages with the greatest initial fitness loss had a larger proportion of innovative (as opposed to restorative) omics changes. I'm afraid I was lost here as the number in parenthesis didn't suggest a correlation to me. It would be better to plot this result on a figure. Btw, the authors report a very strong correlation coefficient ($R=0.99$), but it is based on only 5 data points. If it was not statistically

significant then I wouldn't report the R value, just show the result as a suggestive pattern that remains to be confirmed by future works.

6. Lines #183-185: This sentence should be clarified. What was not possible before this study design? Which conclusions couldn't be made by earlier studies (please cite some specific papers with a similar aim)?

7. Lines #186-187, section header: 'suboptimal pathway usage' sounds as a very term. The section actually has a more specific message: changed flux distribution is more prevalent during compensation than changed flux capacity. Note that the reader may interpret both types of changes as evidence for suboptimality. Btw, it should be made more clear in the text that these two types of changes refer to uKO - eKO comparisons.

8. Fig 4F: a color code of the flux values would be helpful.

9. Line #215: 'Strong evidence for changed TF activation...' Here it would be helpful to precisely define what the authors mean by strong evidence. Statistically significant agreement between metabolome data and expression of specific TUs?

10. Lines #218-220: The activation profiles of 15 TFs changed across all lineages and therefore appear to be non-specific responses. This is worrying as these changes might have nothing to do with compensatory adaptation per se (see my comment #1). At the very least, this issue should be discussed.

11. The section on 'Multiple and competing layers of regulation...' are overly descriptive and hard to follow (especially lines #283-301). Many individual examples are listed without giving enough context / justification and no figure is provided. Why these cases were selected? Why should the reader care?

12. Lines #335 onwards: the authors suggest that many mutations affected global regulators. I wonder if any direct evidence is available (either in the literature or through computational analyses) supporting the impact of some of these mutations? For example, how do we know that the mutations in galR would negate repression of galR controlled operons? Are these loss of function mutations? Without such supportive info, this section reads as a long list of interesting hypotheses.

13. Apparently, my previous question #6 was not addressed at all. It might have been missed.

Reviewer #2 (Remarks to the Author):

The authors have made a significant effort at addressing the concerns of the reviewers and have substantially addressed the majority of my concerns and answered my queries. In this

version, I have several comments and queries, presented in their order of appearance in the manuscript.

- 1) Line 43. It would probably be a good idea to define "fluxomics" either here, or sooner than it is currently explained.
- 2) Lines 47-48. The use of the words "local" and "distal," as well as the identification later of "global" vs. "local" need to be carefully defined. The reader is not going to know what is meant by "distal regulatory changes." If some things are distal, it implies other are proximal. Please define.
- 3) Line 65. For submitted papers, please indicate authors, so the reader can keep an eye out for these upcoming publications.
- 4) Line 69. With respect to "regaining optimality" or the "pursuit of optimality." I brought this up in the previous review as well. First of all, please define "optimal." Is the return to the original state the definition of optimality? Further, it is formally possible that "just good enough" is enough to restore original activity. For example, in the unevolved strain Compound X is at a particular concentration, which we know directly contributes to the maximum growth rate. In the KO strain, the level of X is significantly reduced to the point where the cell now grows more slowly. A mutation in the evolved strain restores levels of X to 10% of the original concentration and now the cell grows at the maximum rate. So, what is the optimal amount of X? The original 100% or the 10%? Frequently, it seems that "original" is the optimal, but I don't think that's what the authors mean... the authors just need to carefully define terms.
- 5) For all of the genes/pathways described in the paragraph starting at Line 88, please add a bit of text describing just what these enzymes or pathways are involved in...what processes?
- 6) Line 92. Comma needed after "... EIIA, respectively)
- 7) Line 118. In addition to the minimum numbers, I would like to see the average numbers as well. Similarly, please quantify the mutational changes.
- 8) Line 119. The authors refer here and elsewhere in the paper to "genomic mutations." What other kinds of mutations could they be? I think just "mutations" is sufficient.
- 9) Line 124. This came up in the first version of the paper as well. The authors keep referring to "the first two modes of the data." This implies that there are more. Since those other modes are not discussed, perhaps just refer to "two modes of the data?"
- 10) Line 127. The authors refer here to "recovery of the reference state." With respect to Comment #4, is this different than optimization?
- 11) Line 142. Isn't the generation of diversity directly linked to the "drive towards fitness?" Are the mutations being selected due to the imperative to increase fitness? I'm not sure if these two concepts are being appropriately described here.
- 12) Line 144. Please define the six profiles and discuss just what they mean. They are not defined in the main text or the figure legend. Also, please describe how the six were chosen.
- 13) Line 196. Please define ED. (Entner-Doudoroff?)
- 14) Line 213. Define TF here, not done until line 215.
- 15) Line 214. Define TU here, not done until line 231.
- 16) Line 217. First use of term "local" to describe some transcription factors vs. global transcription factors. This is not a use of the term "local" that I am familiar with. As stated

above, it needs to be defined and defined clearly. Along with support that this is a standard term of some kind. I think a more common usage is that global transcription factors can affect many unrelated pathways and specific transcription factors affect related pathways. Just be clear!

17) Line 228. Please clarify what is meant by "of their regions..." And also describe the significance of these being a mutation in rpoB.

18) Line 247. Should that read as "L-Tyr?"

19) Line 248. How is the "regulatory interaction" between TyrR and aroF measured, since isn't the output the expression of aroF? What's the difference? Please clarify.

20) Line 296. Please describe Nac better. This description is incomplete.

21) Line 301. Does ppGpp act as a direct regulator of transcription? I thought it primarily acted through translational control. Please verify.

22) Line 318. Should this read "...by RcsA and RcsB." Not RcsAB?

23) Line 350. Should read "A series of mutationS..."

24) Line 352. Should Lon be capitalized?

25) Line 354-355. As written it implies that RpoC is a sigma factor, similar to RpoD. However, RpoC is a component of the core RNA polymerase, not a sigma factor.

26) Line 362. The observation about methylglyoxal is interesting since it is such a potent DNA- and protein-damaging agent. The selection of modulate its production should be strong.

27) Line 369. Capitalize NADPH.

28) In the References: properly italicize all genus and species names, as well as gene names; something is incorrect either in Ref. #18 or #19, since they are the same journal, but the page number style is very different. I think the problem is in #18; Ref. #49 needs page numbers; Ref #69, mBio....lower case "m".

Reviewer #3 (Remarks to the Author):

Overall, the authors have addressed most of the concerns raised, and produced a more readable manuscript. I think the data generated by this work will be valuable for the community, and the hypotheses proposed will be interesting for people to ponder. I still feel that between the hard data and some of the interpretations (e.g. the causal general schema proposed) there are several layers of interpretation, even if, as the authors suggest, convergence of multiple omics data is a helpful strategy to point to potential mechanisms. There is no mention, for example of the importance of the different timescales involved (transcriptional vs. allosteric regulation), or of the possible relevance of mRNA and protein stability and degradation – processes which are unexplored here, and could greatly affect the causal connection between the different changes that happen in response to a gene deletion.

Response to Reviewers

Table of Contents:

Table of Contents:	1
Author's comments:	2
Reviewers' comments:	3
Reviewer #1 (Remarks to the Author):	3
Reviewer #2 (Remarks to the Author):	7
Reviewer #3 (Remarks to the Author):	11
References	13

**Author's comments:**

We would like to thank each of the reviewers again for their time in providing a detailed and
thorough critique of the manuscript. We have addressed each of the reviewer comments to the
best of our abilities. We hope that the reviewers feel their input has led to a much improved
manuscript that will hopefully have a high impact for the scientific community.

Our detailed response to each of the specific comments of each of the reviewers are given in
the sections below.

Reviewers' comments:

Reviewer #1 (Remarks to the Author):

I thank the authors for their significant efforts made towards improving the manuscript. The
revised version is much more focused and improved a lot in clarity. The authors also addressed
most of my previous concerns. However, few issues were not fully addressed and the revision
process raised some new questions as well that should be addressed before publication.

1. Compensatory adaptation is widely defined as a fitness increase that is disproportionately
greater in genetic backgrounds carrying a harmful mutation compared to a control. However, the
authors didn't find a significant fitness increase in the evolved lines of two knockouts.
Interestingly, even these lines show large changes in omics data. Based on these observations,
the authors argue in their response letter that maybe it's time to revise the definition of
compensatory adaptation. I'm not convinced by the latter argument and I doubt many
geneticists / evolutionary biologists would be so.

In my eyes, this discrepancy either represents a lack of evidence for omics changes driven by
compensatory mutations (could neutral mutations be involved?) or lack of resolution to measure
small fitness increases. In either case, the degree of fitness and omics changes are largely
decoupled suggesting that massive network rewiring can take place without any substantial
fitness gain. Obviously, this raises the possibility that some of the omics changes seen in the
other evolved lines, where fitness improved, might be irrelevant for compensation. For this
reason, the issue of lack of fitness compensation should be openly discussed (mentioned in the
results section and further interpreted in the conclusion).

We agree with the reviewer that the “decoupling” of degree of fitness change and degree of
omics data change as shown in the *gnd* and *sdhCB* evolved lines is an important point that
should be discussed further. In line with the reviewer's suggestion, we have expanded our
discussion in the results to discuss the lack of observed fitness compensation in the background
of large omics shifts. The new discussion has been added as the second to last paragraph in
the section “Component profiles reveal systematic variations between ALE lineages, KOs, and
measured data”.

2. Open issues and caveats are not discussed at all in the conclusion / discussion. This would
be important as most conclusions drawn about molecular mechanisms are based on correlative
evidences instead of experimental manipulations. Clearly, the large amount of omics data
cannot substitute for experiments designed to prove specific hypotheses about causality. In this
sense, there is a room for future works on demonstrating the causality of specific mutations,
metabolite concentration changes, etc. This should be discussed upfront.

The reviewer is absolutely correct that the systems biology study presented in this manuscript
does not focus on the time, resource, and labor intensive process of validating molecular
mechanisms that were highlighted or discovered via the bioinformatics approaches used.
Instead, many of the bioinformatics approaches we use recapitulate well known regulatory
mechanisms (in order to demonstrate that they are accurate), but also provide hypotheses for
new lines of inquiry. We provide several supplemental tables (Table S5-6) that organize and
present all of our hypothesis for future experimental validation.

However, as noted by the reviewer, our analyses do not substitute for well designed
experimental follow ups to validate the hypothesized mechanisms. We have made this more
explicit in the introduction to make the reader aware right away that this is a systems biology
study that is intended to reveal insight on a broader scale that includes well founded hypotheses
for future experimental validation. We have also made a call to readers by referencing
supplemental Tables S5-6 and noting the identified mechanisms that would be fruitful targets for
experimental validation.

3. Intro, line 54: many of the references cited here (refs# 10-23) are not about compensatory
evolution. I suggest to keep only the directly relevant citations to avoid confusion.

The reviewer is correct that many of the citations on what was line 54 were not directly about
compensatory evolution. The citations were meant to reference the general experimental
scheme of utilizing ALE to better understand gene loss. We have reworded this sentence to
make this more clear. The subsequent sentences that refer only to refs# 19-21 and 23 were
intended to be specific to compensatory evolution.

4. I couldn't retrieve the list of specific mutations in each genetic background. The readers will
surely be interested in this piece of information.

We regret this oversight. The supplemental table was removed during the revision. We have
reinstated the table as Table S11 that includes a list of all mutations in each of the ALE
endpoints as determined by DNA resequencing to make the information more accessible to the
reader.

5. Lines #163-164: here the authors report that lineages with the greatest initial fitness loss had
a larger proportion of innovative (as opposed to restorative) omics changes. I'm afraid I was lost
here as the number in parenthesis didn't suggest a correlation to me. It would be better to plot
this result on a figure. Btw, the authors report a very strong correlation coefficient ($R=0.99$), but
it is based on only 5 data points. If it was not statistically significant then I wouldn't report the R
value, just show the result as a suggestive pattern that remains to be confirmed by future works.

We would like to thank the reviewer for pointing out the confusion in this analysis. We
discovered a typo in the numbers in the text corresponding to the average number of transcript
levels found for each of the lineages. This was due to a change in the ordering of the lineages
from gnd, pgi, ptsHlcr, sdhCB, tpiA to gnd sdhCB, pgi, ptsHlcr, tpiA. The ordering for the

average number of transcripts was not updated in the text, but was updated in the scripts used
to calculate the correlation coefficients.

We have added in the significance of the correlation coefficient as well as a plot of the data (Fig
S1). It is significant with a p-value < 0.05. Note that the Pearson correlation coefficient was
0.94 and not the reported 0.99. However, the reviewer is correct that regardless of significance,
5 observations is only enough to suggest a pattern that needs to be affirmed by future work.
We have noted this in the text as well.

6. Lines #183-185: This sentence should be clarified. What was not possible before this study
design? Which conclusions couldn't be made by earlier studies (please cite some specific
papers with a similar aim)?

We have removed this sentence. This sentence was in a previous draft where the section in
question was combined with the section "Reference strain evolution confirmed the experimental
design".

7. Lines #186-187, section header: 'suboptimal pathway usage' sounds as a very term. The
section actually has a more specific message: changed flux distribution is more prevalent during
compensation than changed flux capacity. Note that the reader may interpret both types of
changes as evidence for suboptimality. Btw, it should be made more clear in the text that these
two types of changes refer to uKO - eKO comparisons.

While not suggested explicitly, we have gone ahead and changed the title of the section to
"changed flux distribution is more prevalent during compensation than changed flux capacity."
We have also added additional text to make it more clear that the categories pertain primarily to
a comparison between uKO and eKO strains.

8. Fig 4F: a color code of the flux values would be helpful.

We do actually provide color codes for the flux values shown in the linear heatmaps. However,
based on the comment, it appears that they may be difficult to interpret. We have tried to make
this more clear in the figure, and have also highlighted the use of the color bars in the figure
caption.

9. Line #215: 'Strong evidence for changed TF activation...' Here it would be helpful to precisely
define what the authors mean by strong evidence. Statistically significant agreement between
metabolome data and expression of specific TUs?

We have added in the definition for what it means to have "Strong evidence", and have also
added a citation to the supplemental methods for an expanded definition for readers who are
interested to know more details or interested in reproducing the analysis.

10. Lines #218-220: The activation profiles of 15 TFs changed across all lineages and therefore
appear to be non-specific responses. This is worrying as these changes might have nothing to
do with compensatory adaptation per se (see my comment #1). At the very least, this issue
should be discussed.

An important finding of this paper is that large portions of regulatory networks that regulate the
majority of metabolism appear to be governed by a key set of metabolites. This observation has
also been noted in other recent studies e.g., (Kochanowski et al. 2017). In line with the
reviewers inquiry, it does appear that the majority of initial responses in the uKO are generic
stress responses based on a “wobbled” metabolome that are at the very least unproductive to
fitness compensation and arguably counter-productive to fitness compensation (e.g., the
example of g6p cycling in uPgi). This is also shown in Figure 2 where the primary response
between the Ref, uKO, and eKO are restorative changes that correlate to a dampening of these
generic stress responses.

At this point, we are quite cautious about expanding on this point too much. We have received
quite a bit of criticism from previous reviewers that arguments about the “counter productive”
nature of induced regulatory circuits in the uKO goes beyond evidence based analysis. We
have, however, taken the liberty to “cautiously” add in this discussion where appropriate, and to
highlight the non-specificity of these responses, and how such non-specific responses could be
counter productive to compensatory adaptation. Specifically, text addressing this issue can be
found in the last paragraph of the section “Perturbed metabolite levels triggered transcription
regulatory network responses in uKOs”.

11. The section on 'Multiple and competing layers of regulation...' are overly descriptive and
hard to follow (especially lines #283-301). Many individual examples are listed without giving
enough context / justification and no figure is provided. Why these cases were selected? Why
should the reader care?

The examples in this section follow those found in Figure 5 and the data given in Table S6. We
have added in the missing references for Figure 5 to this section. We have also expanded the
conclusion of this section to better explain the need for these examples, and more importantly,
why the reader should care about this section at all.

12. Lines #335 onwards: the authors suggest that many mutations affected global regulators. I
wonder if any direct evidence is available (either in the literature or through computational
analyses) supporting the impact of some of these mutations? For example, how do we know
that the mutations in galR would negate repression of galR controlled operons? Are these loss
of function mutations? Without such supportive info, this section reads as a long list of
interesting hypotheses.

Our evidence for making such claims is based on three lines of evidence: 1) structural analysis,
2) bioinformatics to infer how the mutation changes the sequence of the resulting peptide, 3)
gene expression changes in the operons controlled by the regulator.

It is important to note that we decided not to expand upon the details and mechanisms of each
of the mutations found in all of the endpoints in this manuscript. We felt that adding too much
detail would convolute the already complicated narrative. As stated in the introduction, the
details for all eKO mutation analyses were split into separate manuscripts that are currently
under review or submitted to another journal. We have added an alert to the reader that
additional mechanistic details on the individual mutations described can be found in separate
manuscripts.

13. Apparently, my previous question #6 was not addressed at all. It might have been missed.

We apologize for this oversight. In short, to answer our own question that was left in the
response letter, this analysis was done. In the section titled “Evolution to optimal fitness after
gene KO was captured in the first two modes of the data”, the third sentence down starting at
“For almost all cases analyzed...” the highlighted statistic of 74% was based on comparing the
distance of the Ref, uKO, and eKO in the reduced dimensional space shown in the plots in
Figure 2.

Reviewer #2 (Remarks to the Author):

The authors have made a significant effort at addressing the concerns of the reviewers and
have substantially addressed the majority of my concerns and answered my queries. In this
version, I have several comments and queries, presented in their order of appearance in the
manuscript.

1) Line 43. It would probably be a good idea to define “fluxomics” either here, or sooner than it is
currently explained.

We have added in a short, but descriptive, definition of fluxomics right after the use of the term.

2) Lines 47-48. The use of the words “local” and “distal,” as well as the identification later of
“global” vs. “local” need to be carefully defined. The reader is not going to know what is meant
by “distal regulatory changes.” If some things are distal, it implies other are proximal. Please
define.

We would like to thank the reviewer for identifying our confusing use of the words local and
distal. We have edited the text to use instead proximal or pathway specific where the term
“local” was previously used. We have also expanded the clause “distal regulatory changes” in
the introduction to better explain our reference to regulatory shifts that were not in relative close
proximity to the gene KO.

3) Line 65. For submitted papers, please indicate authors, so the reader can keep an eye out for
these upcoming publications.

We have added the author details to the upcoming publications list.

4) Line 69. With respect to “regaining optimality” or the “pursuit of optimality.” I brought this up in the previous review as well. First of all, please define “optimal.” Is the return to the original state the definition of optimality? Further, it is formally possible that “just good enough” is enough to restore original activity. For example, in the unevolved strain Compound X is at a particular concentration, which we know directly contributes to the maximum growth rate. In the KO strain, the level of X is significantly reduced to the point where the cell now grows more slowly. A mutation in the evolved strain restores levels of X to 10% of the original concentration and now the cell grows at the maximum rate. So, what is the optimal amount of X? The original 100% or the 10%? Frequently, it seems that “original” is the optimal, but I don’t think that’s what the authors mean... the authors just need to carefully define terms.

As pointed out by the reviewer, the original state is not necessarily what we mean by “optimal”. Instead, “optimal” indicates the biochemical state that allows for the maximal growth rate that the organism can achieve given the current environmental and genetic conditions. We have added this definition to the introduction after our first use of the term “optimal”. In addition, we have also qualified the term “optimal” in the text by adding the word “fitness” or “growth” afterwards to indicate our intended definition for “optimal”.

5) For all of the genes/pathways described in the paragraph starting at Line 88, please add a bit of text describing just what these enzymes or pathways are involved in....what processes?

We have added a concise, one sentence overview of the enzymatic process that the enzymes carry out, as well as contextual details of the pathway that they function in or the purpose that they serve in metabolism.

6) Line 92. Comma needed after “.... EIIA, respectively)

We have fixed this mistake.

7) Line 118. In addition to the minimum numbers, I would like to see the average numbers as well. Similarly, please quantify the mutational changes.

We have added in the average numbers for the metabolomics, transcriptomics, and phenomics in the eKOs. We have also added in the average numbers for the mutational changes in eRef and the other eKOs.

8) Line 119. The authors refer here and elsewhere in the paper to “genomic mutations.” What other kinds of mutations could they be? I think just “mutations” is sufficient.

We have removed the word “genomic” in the clauses “genomic mutations”

9) Line 124. This came up in the first version of the paper as well. The authors keep referring to
“the first two modes of the data.” This implies that there are more. Since those other modes are
not discussed, perhaps just refer to “two modes of the data?”

In PLS-DA analysis, the modes are rank ordered in accordance with the explanatory power. By
stating the “first two modes of the data” we are referencing and emphasizing that the below
analyses are based off of the two most explanatory modes in the data. It is a common practice
in many decomposition methods to utilize only the most explanatory modes for subsequent
visualization and analysis. We have made this explicit by adding in “most explanatory” after
“first” or “second” and before “mode”.

10) Line 127. The authors refer here to “recovery of the reference state.” With respect to
Comment #4, is this different than optimization?

The use of “reference” was not meant to indicate something other than the optimal state.
Therefore, we have replaced “reference” with “optimal”.

11) Line 142. Isn’t the generation of diversity directly linked to the “drive towards fitness?” Are
the mutations being selected due to the imperative to increase fitness? I’m not sure if these two
concepts are being appropriately described here.

We have added in a clarifying sentence to better explain the relationship between these two
concepts.

12) Line 144. Please define the six profiles and discuss just what they mean. They are no
defined in the main text or the figure legend. Also, please describe how the six were chosen.

We have re-written the caption for Fig. 3 to include a definition for each of the sizes and the
rationale for why those six profiles were chosen.

13) Line 196. Please define ED. (Entner-Douderoff?)

We have updated the acronym

14) Line 213. Define TF here, not done until line 215.

We have updated the acronym

15) Line 214. Define TU here, not done until line 231.

We have updated the acronym

16) Line 217. First use of term “local” to describe some transcription factors vs. global
transcription factors. This is not a use of the term “local” that I am familiar with. As stated above,

it needs to be defined and defined clearly. Along with support that this is a standard term of
some kind. I think a more common usage is that global transcription factors can affect many
unrelated pathways and specific transcription factors affect related pathways. Just be clear!

Following the reviewers suggestion, we have removed “local” and replaced it with “pathway
specific”.

17) Line 228. Please clarify what is meant by “of their regions...” And also describe the
significance of these being a mutation in *rpoB*.

We have rephrased the clause “of their regions...” with “sigma factor DNA binding operons” to
be more explicit. We have also added some explanatory and background text on the reference
to the *rpoB* citation.

18) Line 247. Should that read as “L-Tyr?”

Yes, the reviewer is correct that an error was made in our abbreviation. We have fixed it.

19) Line 248. How is the “regulatory interaction” between TyrR and *aroF* measured, since isn’t
the output the expression of *aroF*? What’s the difference? Please clarify.

We have added some additional text to better explain what we mean by “regulatory interaction”
that should make this statement more clear.

20) Line 296. Please describe Nac better. This description is incomplete.

Nac is a transcription factor that is involved in regulating nitrogen metabolism. We have added
in some additional text to better explain the Nac and its role in nitrogen metabolism.

21) Line 301. Does ppGpp act as a direct regulator of transcription? I thought it primarily acted
through translational control. Please verify.

The reviewer is correct that ppGpp does act through transcriptional control. For the example
given, ppGpp inhibits binding of the RNA polymerase, and hence, gene expression (Donahue
and Turnbough 1990).

22) Line 318. Should this read “...by RcsA and RcsB.” Not RcsAB?

“RcsAB” is correct.

23) Line 350. Should read “A series of mutationS....”

Thank you for identifying the missing “S”. This has been corrected.

24) Line 352. Should Lon be capitalized?

We have corrected the mistake in capitalization.

25) Line 354-355. As written it implies that RpoC is a sigma factor, similar to RpoD. However,
RpoC is a component of the core RNA polymerase, not a sigma factor.

We have revised the sentence.

26) Line 362. The observation about methylglyoxal is interesting since it is such a potent DNA-
and protein-damaging agent. The selection of modulate its production should be strong.

We did find that the selection to rapidly detox methylglyoxal was strong. We found that the
majority of the eTpiA strains shared the same or similar mutations.

27) Line 369. Capitalize NADPH.

We have corrected this mistake.

28) In the References: properly italicize all genus and species names, as well as gene names;
something is incorrect either in Ref. #18 or #19, since they are the same journal, but the page
number style is very different. I think the problem is in #18; Ref. #49 needs page numbers; Ref
#69, mBio....lower case "m".

We have updated and reformatted our references. For some reason, our reference manager
was having problems with those specific citations. We updated them by hand.

Reviewer #3 (Remarks to the Author):

Overall, the authors have addressed most of the concerns raised, and produced a more
readable manuscript. I think the data generated by this work will be valuable for the community,
and the hypotheses proposed will be interesting for people to ponder. I still feel that between the
hard data and some of the interpretations (e.g. the causal general schema proposed) there are
several layers of interpretation, even if, as the authors suggest, convergence of multiple omics
data is a helpful strategy to point to potential mechanisms. There is no mention, for example of
the importance of the different timescales involved (transcriptional vs. allosteric regulation), or of
the possible relevance of mRNA and protein stability and degradation – processes which are
unexplored here, and could greatly affect the causal connection between the different changes
that happen in response to a gene deletion.

We would like to thank the reviewer highlighting the different time-scales of regulatory events as
well as macromolecule stability that were not discussed that could potentially have a large
impact in the observed physiology. We have noted this shortcoming in the conclusions. In

combination with our detailed list of hypotheses generated by this study, a note of this sort
should help guide up and coming researchers where potentially ripe opportunities for fruitful
experimentation lay.

**References**

Donahue, J. P., and C. L. Turnbough Jr. 1990. "Characterization of Transcriptional Initiation
from Promoters P1 and P2 of the pyrBI Operon of Escherichia Coli K12." *The Journal of*
*Biological Chemistry* 265 (31): 19091–99.

Kochanowski, Karl, Luca Gerosa, Simon F. Brunner, Dimitris Christodoulou, Yaroslav V.
Nikolaev, and Uwe Sauer. 2017. "Few Regulatory Metabolites Coordinate Expression of
Central Metabolic Genes in Escherichia Coli." *Molecular Systems Biology* 13 (1): 903.

REVIEWERS' COMMENTS:

Reviewer #1 (Remarks to the Author):

The authors addressed all the points raised in my previous report and the manuscript should be acceptable for publication.